# Climate-induced forest dieback drives compositional changes in insect communities that are more pronounced for rare species

Lucas Sire [1✉], Paul Schmidt Yáñez[2], Cai Wang[3,4], Annie Bézier [1], Béatrice Courtial[5], Jérémy Cours[6], Diego Fontaneto [7], Laurent Larrieu[8,9], Christophe Bouget[6], Simon Thorn [10], Jörg Müller [10,11], Douglas W. Yu [3,12], Michael T. Monaghan [2,13], Elisabeth A. Herniou[1] & Carlos Lopez-Vaamonde [1,5]

Species richness, abundance and biomass of insects have recently undergone marked declines in Europe. We metabarcoded 211 Malaise-trap samples to investigate whether drought-induced forest dieback and subsequent salvage logging had an impact on ca. 3000 species of flying insects in silver fir Pyrenean forests. While forest dieback had no measurable impact on species richness, there were significant changes in community composition that were consistent with those observed during natural forest succession. Importantly, most observed changes were driven by rare species. Variation was explained primarily by canopy openness at the local scale, and the tree-related microhabitat diversity and deadwood amount at landscape scales. The levels of salvage logging in our study did not explain compositional changes. We conclude that forest dieback drives changes in species assemblages that mimic natural forest succession, and markedly increases the risk of catastrophic loss of rare species through homogenization of environmental conditions.

[1] Institut de Recherche sur la Biologie de l'Insecte (IRBI), UMR 7261, CNRS-Université de Tours, Tours, France. [2] Leibniz Institute of Freshwater Ecology and Inland Fisheries (IGB), Müggelseedamm 301, 12587 Berlin, Germany. [3] State Key Laboratory of Genetic Resources and Evolution, Kunming Institute of Zoology, Chinese Academy of Sciences, Kunming, Yunnan 650223, China. [4] Kunming College of Life Sciences, University of Chinese Academy of Sciences, Kunming, China. [5] INRAE, Zoologie Forestière, F-45075 Orléans, France. [6] INRAE 'Forest Ecosystems' Research Unit – Biodiversity team Domaine des Barres, F-45290 Nogent-sur-Vernisson, France. [7] Water Research Institute, National Research Council of Italy, CNR-IRSA, Largo Tonolli 50, 28922 Verbania Pallanza, Italy. [8] Université de Toulouse, INRAE, UMR DYNAFOR, Castanet-Tolosan, France. [9] CRPF Occitanie, Tarbes, France. [10] Field Station Fabrikschleichach, Department of Animal Ecology and Tropical Biology, Biocenter, University of Würzburg, Glashüttenstraße 5, 96181 Rauhenebrach, Germany. [11] Bavarian Forest National Park, Freyunger Str. 2, 94481 Grafenau, Germany. [12] School of Biological Sciences, University of East Anglia, Norwich Research Park, Norwich, Norfolk NR47TJ, UK. [13] Institut für Biologie, Freie Universität Berlin, Königin-Luise-Straße. 1-3, 12489 Berlin, Germany. ✉email: lucas.sire@univ-tours.fr

Insects are vital components of biodiversity, providing important ecosystem services such as pollination and pest regulation, while also performing disservices as disease vectors and plant pests[1]. Global changes, including those of climate, land use and land cover, can lead to degradation and habitat loss, chemical and light pollution or invasive species. These changes have caused major decreases in biomass, abundance and species richness of insects[2], and this accelerating decline has become a major cause for concern. Yet, understanding the relative impacts of these drivers underlying insect decline and its propensity is a complex task[3].

Forests are thought to act as buffer zones from rapid anthropogenic changes in adjacent areas, and contribute to biodiversity conservation by providing refuge for rich insect communities[4]. However, forests are increasingly suffering from climate-induced tree dieback, regardless of protection measures[5], which could have long-term consequences for forest biodiversity. Tree diebacks largely result from more frequent, intense and longer droughts, especially in Europe[6]. Indeed, severe summer droughts can induce tree mortality in particular if they trigger insect pests and pathogen outbreaks. While heatwaves can negatively impact some insect pest species or pathogens by imposing heat stress, others can benefit from it, favouring distribution range expansion and population outbreaks, which can cause massive forest diebacks[7]. Bioclimatic models predict that the extent, severity and duration of droughts will increase as a result of climate change, notably in temperate and Mediterranean areas[8].

Drought-induced forest diebacks can cause major structural changes such as an increase in canopy openness and reduction in foliage density, which in turn increases light availability, potentially changing the community structure of understory plants and their associated herbivorous insects[9]. While insect decline is independent of forest protection status, it has been presented as lower in terms of species richness for plots undergoing dieback[2]. Indeed, tree dieback can also increase the availability of resources stored in vital trees (i.e. deadwood), sunlight and microhabitats, increasing species richness for multiple insect groups including bees, wasps, hoverflies, saproxylic beetles, as well as multiple red-listed insect taxa following insect pest outbreaks[10], or canopy-dwelling beetles in declining oak forests[9]. However, no significant change in species richness of ground-dwelling carabids and spiders has been associated with climate-induced dieback in a beech-dominated forest[11]. Meanwhile, forest management practices, like salvage logging (i.e. cutting down dying trees to salvage their timber value), can also have contrasting effects on biodiversity[12]. For instance, species richness of saproxylic insects increased with bark beetle outbreaks[10], but decreased with deadwood harvesting[13]. Even though specific to each of these well-studied taxa previously stated, species richness in general appears to be boosted by tree diebacks, while salvage logging seems to have no effect overall[14].

The identification of samples down to species level is important in ecological studies because adaptive response to disturbances can be highly species-specific[15]. Unfortunately, studies on the response of insects to forest disturbance often focus on a few well-known taxa and exclude important hyper-diverse groups such as Diptera and Hymenoptera that are difficult to identify to species level[9,10] and as a result leave changes undetected. In addition, studies on the response of insect biodiversity to forest disturbances are based on species richness, which has been used as a surrogate for ecosystem functionality[16]. However, richness alone is often a poor indicator of biodiversity change compared to ecological guilds within the community[17]. Quantifying these changes requires high taxonomic resolution that is often not available. The morphological identification that richness measures are typically derived from are often limited to well-known insect bioindicators that are overrepresented in the literature, hence biasing species richness per se and impeding further holistic community-based and ecological studies. Unfortunately, accurate species identification of hyper-diverse insect groups such as dipterans and hymenopterans is difficult because of the taxonomic impediment[18], often hampering—or limiting to few remarkable groups—the study of these species-rich taxa despite their ecological importance[19]. The use of DNA barcoding as a tool for species identification[20] now allows bypassing this taxonomic impediment[21]. As exemplified by Wang et al.[22] targeting poorly described Chinese entomofauna, the use of DNA metabarcoding allows researchers to tackle more comprehensive insect biodiversity studies and facilitates the documenting of all trophic guilds and their responses to forest disturbances. The authors suggested that forest diebacks induced by bark beetle outbreaks could drive a transition from homogenous plantations to more biodiversity-friendly heterogenous forests[22].

To measure the response of insects to forest dieback and subsequent salvage logging, there is a need to sample as many taxa as possible to detect any possible variation among different taxa and functional groups. For instance, floricolous species are expected to increase in diversity and abundance following tree dieback due to greater canopy openness[15]. Similarly, parasitoid wasps, which are rarely included in this kind of ecological study, may benefit from temperature rises and associated tree diebacks, because the more complex a forest is the more diverse the parasitoid communities are[23]. Parasitoids may also suffer from poorer host quality, especially of sap-feeders directly affected by tree health under drought conditions[24]. Forests undergoing diebacks are thus expected to induce guild-specific responses, which may have contrasting effects on species richness of each insect group depending on their respective ecology[9]. A comprehensive approach to sampling and identification of taxa is therefore needed to improve our understanding of the effects of forest disturbances on total insect biodiversity.

Here, we study insect diversity in montane Pyrenean forests dominated by silver fir (*Abies alba* Mill), a conifer species sensitive to drought[25] that suffered severe climate-induced diebacks due to several heatwaves that have occurred since 2003[26]. Current forest management practices often implement salvage logging at the first signs of dieback[26]. Despite the high conservation value of Pyrenean silver fir forests, the effects of dieback and salvage logging on associated insect fauna remain unknown. To address this gap, we studied the communities of aerial insects over 56 natural (i.e. non-experimentally modified) silver fir forest plots (Fig. 1) that varied in dieback level and management practices. From late spring to early autumn 2017, insects were mass-trapped monthly using Malaise traps, resulting in 222 samples that we analysed using DNA metabarcoding of a fragment of the mitochondrial COI gene region. Based on BOLD DNA reference libraries[27], the recovered Molecular Operational Taxonomic Units (MOTUs) were taxonomically identified. Ecological functions (i.e. floricolous and parasitoid insects) were attributed using family-level information from published literature whenever possible. We used generalized linear models to assess whether dieback levels and salvage logging influence the structure and diversity of forest insect communities and functional guilds.

We hypothesized that species richness would remain stable based on similar amounts of species gain and loss throughout the levels of forest dieback and salvage logging, but that changes in species composition of local insect communities would occur across sites with different disturbances. We also hypothesized that functional guilds would respond differently to forest dieback intensity; in particular the diversity of parasitoid and floricolous insects in areas of high forest dieback with greater canopy openness. As expected, we found no change in species richness

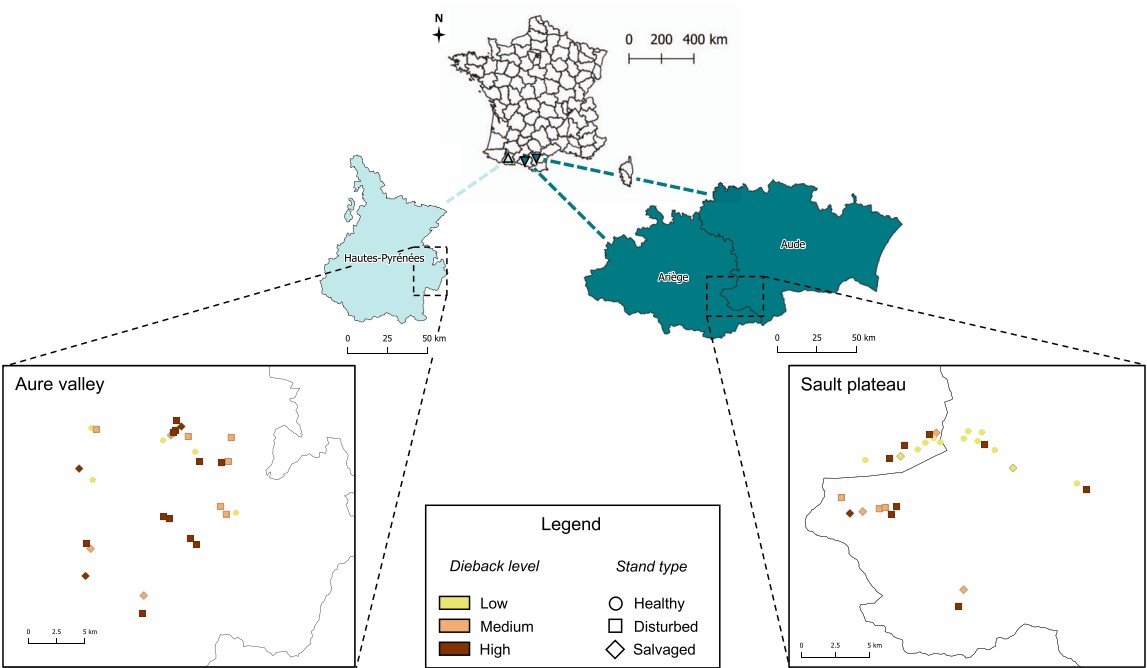

**Fig. 1 Map of the study area with sampled plot description.** Map of study area across the three concerned administrative regions of the French Pyrenees. Sampled districts are highlighted in shades of blue with plots coloured and shaped according to dieback level and stand type, respectively. Dieback level was assessed using ARCHI method and stand type using IBP (Index of Biodiversity Potential) measurements.

but variations in community compositions across dieback levels and management practices. We performed zeta order analyses[28,29] to further investigate the dataset structure and identify the nature of these community changes, as well as the species compositional and functional turnover. We also used zeta analyses in multi-site generalized dissimilarity modelling (zeta.msgdm)[30] to study the impact of environmental features (i.e. geographic distance to the nearest plot, altitude, canopy openness, total amount of deadwood, basal area, tree diversity, density of very large trees and both Tree-related Microhabitats (TreM) diversity and density) in driving these community structural changes and further discuss their associated consequences for forest management and conservation. Finally, we used the fourth-corner model[31] to highlight hypothesized winning and losing insects to forest disturbances as well as idiosyncratic responses of insect orders and functional groups.

## Results

**Diverse insect community with high temporal and spatial turnover.** A total of 59,321,436 raw sequencing reads (individual forward and reverse reads) were obtained from the sequencing, and 11,802,769 sequences were recovered after complete demultiplexing and data processing including applying our criteria of at least 3 reads present in each of the 3 PCR replicates for each retained MOTU. Overall, the mean number of reads per sample was 55,937 ± 22,120 (SD) (max. 157,012; min. 12,669). Of the 222 Malaise-trap samples studied, 211 samples yielded results after demultiplexing.

Metabarcoding recovered thousands of locally rare species and unnamed "dark taxa" (i.e. species-level 'arthropod' taxa in DNA reference libraries without a species name assigned and/or species-rich taxa, often small in body size and underdescribed)[32]. From the 211 Malaise-trap samples, we recovered 2972 MOTUs (see Supplementary Data 1 for the complete MOTU list), and we estimated a total richness of ~4000 MOTUs with iNEXT extrapolation set on incidence frequency datatype and Hill number $q = 0$ parameter to account for observed

MOTU diversity, with species accumulation curves approaching saturation (Fig. 2a, Supplementary Fig. 1). Our 4-month sampling effort (from late spring to early autumn) thus captured *ca.* 75% of the total Malaise-trappable insect diversity in the sampled areas allowing us to reach ~90% sample coverage (Fig. 2a–c). Applying a threshold of 97% sequence similarity for species-level discrimination and keeping unambiguous taxonomic matches for higher taxonomic ranks, 100%, 84.8%, 75.4% and 52.4% of the total 2972 insect MOTUs were assigned to a total of 15 orders, 258 families, 1193 genera and 1558 species names, respectively (Supplementary Data 1 and 2). Two orders, Diptera and Hymenoptera, together represented 73% of all MOTUs, as expected with Malaise-trapped samples (Fig. 3a), with 641 (49.7%) and 371 MOTUs (41.4%) identified to species, respectively (Supplementary Fig. 3). As with the whole dataset (Fig. 2a), the accumulation curves of the five most represented insect orders approached asymptotes (Fig. 3b).

We observed a high rate of temporal species turnover across months, as evidenced by only 360 MOTUs (12% of the total MOTUs diversity) being detected in all 4 months (i.e. from mid-May to mid-September) (Fig. 2d). Thus, even though our sampling was efficient at recovering high biodiversity, sampling throughout the complete growing season is likely to increase the total diversity occurring in the region (Fig. 2d). After grouping monthly temporal replicates into sample sites, geographic turnover was also observed, with only 45% of the total insect species sampled occurring in both of the sampled districts Aure valley (Central French Pyrenees) and Sault plateau (Eastern French Pyrenees) (Fig. 2a), the remaining 31% and 24% being specific to each, respectively. In addition, these district-specific taxa were the most rarely caught throughout all the different plots of their respective districts (Supplementary Fig. 2a).

**Climate-induced dieback influences the composition but not the species richness of insect communities.** We grouped individual sample sites into three dieback categories (low, medium, high) based on the proportion of drought-affected and dying trees

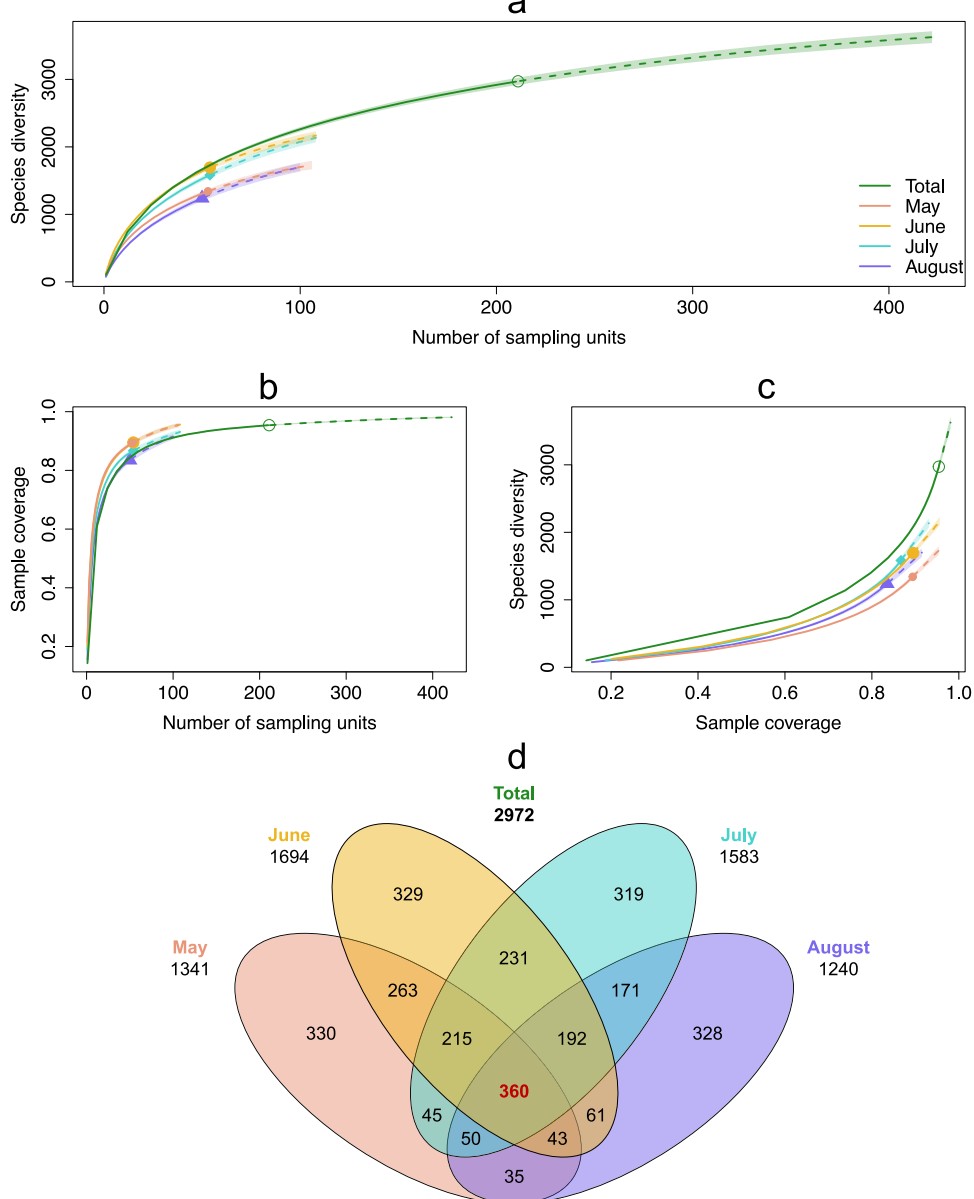

**Fig. 2 iNext species accumulation curves for Malaise-trap samples and temporal turnover. a** Species richness (MOTUs) per sampling unit recovered through metabarcoding of Malaise-trap samples (solid lines). Incidence frequency-based species richness extrapolations (dashed lines) were performed over the total sampling period (from May 15 to September 15, 2017) (green curve) as well as for the following 4-weeks temporal sampling categories: May, June, July, August to assess the potential variations in both sampling effort and species richness (pink, yellow, cyan and violet curves, respectively). Sampling units are the total number of Malaise-trap samples available per temporal category. Shaded area represents 95% confidence interval. **b** Sample coverage (sampling efficiency of sampling units to recover the expected biodiversity) per number of sampling units. **c** Species diversity (MOTUs) per sample coverage. Accumulation curves indicated that the 211 sampling units used in the present study allowed a 90% sampling coverage of the given area and sampling period and allowed us to recover nearly 75% of the total Malaise-trappable diversity. Estimations gave around 4000 trappable species and the need of more than 400 sampling units on-site to recover them. **d** Venn diagram showing the species (MOTUs) turnover in the French Pyrenees across the 4-month sampling period of the study. Total number of MOTUs found every four weeks is given under the name of each starting month. Number of MOTUs shared between months is given within the diagrams. The high MOTU numbers specific to each month and the comparatively low number of shared MOTUs throughout the entire sampling period (highlighted in red) indicated an important temporal turnover of insect species.

for each plot. Insect species richness was compared across the three dieback categories but also across three defined stand types including management practice (healthy, stands expressing dieback but not salvaged—hereafter 'disturbed'—, stands expressing dieback and being salvage logged—hereafter 'salvaged'). No response could be detected in terms of changes in species richness from all insect or functional groups tested, neither for the three levels of forest dieback nor the stand types (Table 1). However, a significant difference in species richness was found between the

two sampled districts for Coleoptera only, with more species of beetles detected in Sault plateau (Table 1).

As the dataset was geographically structured (i.e. Aure valley and Sault plateau districts), we analysed the dataset using the nearest-neighbour (NN) sample-selection scheme (Fig. 4). The NN model fitted better to a power-law function than to an exponential one (Fig. 4c, d; $AIC_{(NN, Exp)} = 0.89$, $AIC_{(NN, PL)} = -122.78$), which was consistent with community assembly being driven by niche differentiation over stochastic assembly. We also

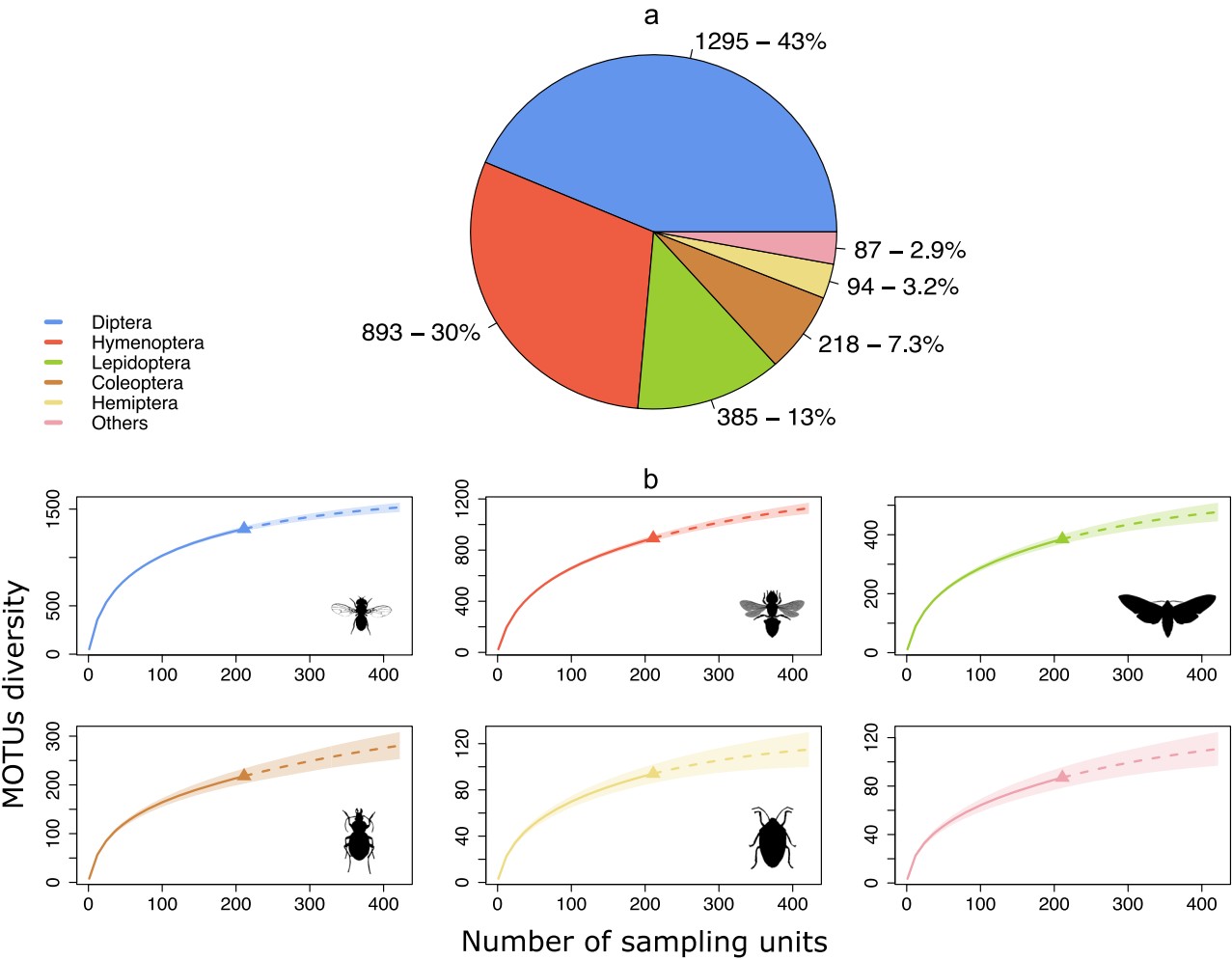

**Fig. 3 Taxonomic composition and Gamma diversity analysis. a** Pie chart of the 15 insect orders sampled showing the taxonomic affiliation and the respective number of MOTUs recovered, as well as their total proportion. For clarity purposes, only the five orders with highest number of species (Diptera, Hymenoptera, Lepidoptera, Coleoptera and Hemiptera) are represented (in blue, red, green, brown and yellow, respectively), with the 10 remaining orders (Blattodea, Ephemeroptera, Mecoptera, Neuroptera, Orthoptera, Plecoptera, Psocodea, Raphidioptera, Thysanoptera and Trichoptera) clustered within the "Others" category in pink. Representativeness of each insect order in the dataset was consistent with the known bias of Malaise-trap sampling, especially toward dipterans and hymenopterans. **b** Incidence frequency-based accumulation curves (solid lines) with species richness extrapolations (dashed lines) for each insect order. Shaded area represents 95% confidence interval. As all accumulation curves were nearly plateauing, this indicated that for each represented insect order, almost all Malaise-trappable diversity over the sampling period had been successfully recovered.

ran a zeta-diversity analysis with the non-geographically structured all-combinations (ALL) sample-selection scheme (Supplementary Fig. 4), and the results were similar but showed a much weaker fit to the power-law function (Supplementary Fig. 4c, d; $AIC_{(ALL,\ Exp)} = 0.63$, $AIC_{(ALL,\ PL)} = -20.97$). The rapid decline in zeta diversity between zeta orders 2 and 10 indicated that compositional turnover was mainly driven by rare species; few species were shared in 10 or more sites (Fig. 4a; Supplementary Fig. 2b). However, re-visualising the decline curve in Fig. 4a as a zeta retention-rate curve (Fig. 4b) showed that the few species that did occur in ≥10 sites ($n = 42$ species) were highly prevalent but only two species were shared by 55 out of 56 total sites. Interestingly, this decline curve also demonstrated the drop of species retention rate at 28 sites (equivalent to the number of sampled plots in each district), accounting for the strong geographic effect that induced very few shared species between Aure valley and Sault plateau. Re-analysis using the ALL sample-selection scheme produced similar results in shared species but smoothed geographic effects (Supplementary Fig. 4).

The impacts of drought-induced forest dieback intensities and stand types on community compositional changes were evaluated

for the total sampling (Fig. 5a). Overall, forest dieback was found to induce significant changes in insect community assemblages for all tested groups but Coleoptera (Supplementary Table 1). However, no significant variation in community composition was found across stand types, hence no effect of salvage logging could be detected (Supplementary Table 1). Regardless of dieback intensity and stand type, each insect community assemblage tested differed significantly between districts (Fig. 5b, Supplementary Table 1). Each dieback category hosted particular sets of species, yet insect communities of low dieback level plots were more similar to each other and more distinct than those of medium and high dieback level plots (Fig. 5a; Supplementary Fig. 2b). Furthermore, taxa specific to a particular dieback level were mostly rare taxa (i.e. taxa with low prevalence) (Supplementary Fig. 2b). Finally, community composition variations found across districts were reflected by Sault plateau plots sharing more species than those of Aure valley plots (Fig. 5b).

**Winners and losers of forest disturbances.** Zeta decline analyses allowed us to assess community assemblages and species

**Table 1 Impact of forest dieback and salvage logging on insect species richness.**

| Studied group | Condition | Degree of freedom | Chi-square | P-value (95% confidence) | Significance |
|---|---|---|---|---|---|
| Total insects | | | | | |
| | Dieback level effect | 2 | 2.110 | 0.3483 | N.S. |
| | Stand type effect | 2 | 0.945 | 0.6230 | N.S. |
| | District effect | 1 | 0.520 | 0.4710 | N.S. |
| Coleoptera | | | | | |
| | Dieback level effect | 2 | 0.357 | 0.8366 | N.S. |
| | Stand type effect | 2 | 0.906 | 0.6357 | N.S. |
| | District effect | 1 | 13.957 | **0.0002** | *** |
| Diptera | | | | | |
| | Dieback level effect | 2 | 1.978 | 0.3719 | N.S. |
| | Stand type effect | 2 | 1.806 | 0.4053 | N.S. |
| | District effect | 1 | 1.815 | 0.1779 | N.S. |
| Hemiptera | | | | | |
| | Dieback level effect | 2 | 0.142 | 0.9315 | N.S. |
| | Stand type effect | 2 | 1.640 | 0.4404 | N.S. |
| | District effect | 1 | 1.587 | 0.2078 | N.S. |
| Hymenoptera | | | | | |
| | Dieback level effect | 2 | 2.724 | 0.2562 | N.S. |
| | Stand type effect | 2 | 1.239 | 0.5384 | N.S. |
| | District effect | 1 | 1.042 | 0.3074 | N.S. |
| Lepidoptera | | | | | |
| | Dieback level effect | 2 | 0.742 | 0.6899 | N.S. |
| | Stand type effect | 2 | 0.150 | 0.9279 | N.S. |
| | District effect | 1 | 0.925 | 0.3361 | N.S. |
| Floricolous | | | | | |
| | Dieback level effect | 2 | 2.330 | 0.3119 | N.S. |
| | Stand type effect | 2 | 0.372 | 0.8301 | N.S. |
| | District effect | 1 | 0.572 | 0.4495 | N.S. |
| Non-floricolous | | | | | |
| | Dieback level effect | 2 | 2.100 | 0.3499 | N.S. |
| | Stand type effect | 2 | 1.341 | 0.5114 | N.S. |
| | District effect | 1 | 0.109 | 0.7415 | N.S. |
| Parasitoids | | | | | |
| | Dieback level effect | 2 | 2.422 | 0.2979 | N.S. |
| | Stand type effect | 2 | 0.798 | 0.6709 | N.S. |
| | District effect | 1 | 3.223 | 0.0073 | N.S. |
| Non-parasitoids | | | | | |
| | Dieback level effect | 2 | 2.018 | 0.3646 | N.S. |
| | Stand type effect | 2 | 1.136 | 0.5666 | N.S. |
| | District effect | 1 | <0.001 | 0.9860 | N.S. |

Generalized linear models for species richness variations of different study groups (i.e. total insects, five individual insect orders, and four functional groups) compared across respective environmental conditions of diebacks (i.e. low, medium and high forest dieback levels) and stand types (i.e. healthy, disturbed and salvaged logged). Functional groups (i.e. Parasitoids/non-parasitoids, floricolous/non-floricolous insects) were assigned using each MOTU's taxonomic family. Significance is given by "***" while "N.S." stands for non-significant.

retention rates across plots for the five main insect orders represented and for functional groups (floricolous/non-floricolous and parasitoid/non-parasitoid species) within each dieback category and stand type. We detected that species were retained differently within both dieback intensity gradient and stand types, regardless of the model scheme used (Figs. 6 and 7; Supplementary Figs. 5 and 6). Both Lepidoptera and Hemiptera showed a rapid drop and a lack of structure in zeta ratio along the dieback gradient (Fig. 6a–c) and between the different stand types (Fig. 7a–c), indicating there were no species shared across zeta order ranges of both environmental gradients. Similar zeta declines were observed at higher zeta orders for Coleoptera at low dieback level and in healthy stands (Figs. 6a and 7a), as well as for Diptera and Hymenoptera at high dieback level and in disturbed stands (Figs. 6c and 7b). These results highlighted that rare species shaped Coleoptera communities mostly at low dieback level, whereas common Coleoptera species were found across plots of higher dieback gradients. Conversely, both Diptera and Hymenoptera species assemblages were less diverse at low dieback level and healthy stands while high dieback level and

disturbed stands favoured complete species turnover with no common species retained (Figs. 6a–c and 7a, b). Regarding functional assemblages, all had common species likely to be found across low and medium dieback gradient, as well as healthy and salvaged forest stands (Figs. 6d–e and 7d–f). Nevertheless, while a similar pattern was observed for non-floricolous and non-parasitoid species assemblages at high dieback level or within disturbed stands, zeta ratio of decline for both parasitoid and floricolous species cohorts fully dropped, indicating species compositional turnover within the two functional groups and no core species shared across 18 to 23 zeta order range (Figs. 6f and 7e). This lack of structure was likely driven by effects of high dieback level and disturbed stands observed on both Diptera and Hymenoptera, which included many taxa of pollinators and/or parasitoids (Figs. 6c and 7b). Overall, while few drops in species retention rates due to geographic effect were noticeable throughout the different environmental conditions (Figs. 6 and 7), main effects of both dieback level and stand types on species retention rates remained visible in ALL model scheme (Supplementary Fig. 5–6).

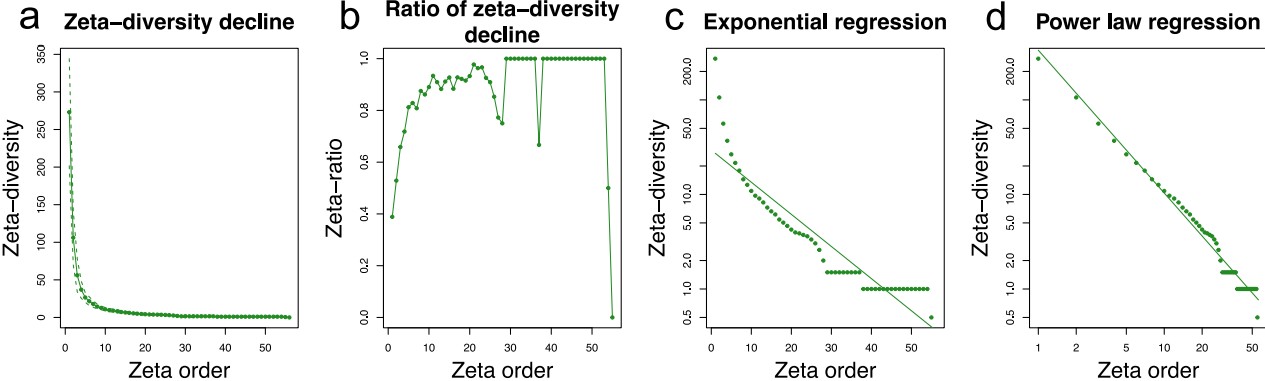

**Fig. 4 Zeta-diversity decline and model fitting on the insect fauna for nearest-neighbour combinations and assemblages.** Zeta-diversity analyses per zeta order (i.e. $\zeta_i$) here referring to plots (from $\zeta 2$ to $\zeta 56$) for two computing schemes. Representations consider model scheme that computes nearest-neighbour plot combinations and assemblages (NN) with parameter sample set to 5000 and Monte–Carlo (mc) sampling. **a, b** Zeta-diversity decline and 0 to 1 scaled ratio of zeta-diversity decline, representing species shared and species retention rate (i.e. the retention probability of common species in the community) across $\zeta_i$, respectively. Zeta-diversity decline curve (**a**) indicated high heterogeneity in insect diversity across the studied plots as fewer and fewer species were shared between 2–9 plots and almost no common species were being shared when 10 plots or more were considered. Zeta ratio (**b**) highlighted the high prevalence of the few species shared across 2–54 plots that were likely to be sampled, and the drop of species retention rate at zeta order $\zeta 28$ corresponded to geographic distance between the two sampled districts. **c, d** zeta-diversity model fitting to exponential and power-law regressions, respectively. Fit to exponential regression implies equal and stochastic turnover among species whereas fit to power-law regression as represented here indicates turnover driven by niche differentiation processes and rare species less likely to be found across the whole dataset.

Winning and losing insect orders in terms of prevalence over dieback gradient and stand types were assessed using a "fourth corner" modelling. We found a higher prevalence in low dieback level stands and conversely, a detrimental effect of salvage logging over Coleoptera (Fig. 8). Furthermore, the lack of a decrease in species retention rate for both Hymenoptera and Diptera at low dieback level or in healthy stands (Figs. 6a and 7a) could be explained by a lower prevalence of these two orders in these environments (Fig. 8). Interestingly, Hymenoptera but not Diptera were winning from a particular level of dieback, especially with a higher prevalence at medium dieback level (Fig. 8a), while both Lepidoptera and Hemiptera diversity were favoured by stand disturbances in general (Fig. 8b).

Finally, by analysing congruences between MOTUs and dieback gradients using IndVal analyses, we highlighted species-specific responses to forest dieback. We significantly associated MOTUs to low and medium but not to high dieback levels (Supplementary Table 2). These MOTUs could be linked to specific forest dieback conditions with particular environmental niches and therefore be considered losing over the general dieback gradient. The remaining species may either be too poorly sampled across plots to assess significant linkage or likely to be spread across multiple levels of dieback.

**Environmental drivers of species turnover.** When assessing the contribution of our eight variables as drivers of the compositional turnover across zeta orders, we found that distance between plots played an important role in explaining the observed variance, especially in a two-plots to 10-plots comparison—hence dominated by rare species—and even greater at zeta orders above 28, thus on common species (Fig. 9a). This increase in geographic effect was in accordance with the relative number of plots in each respective district, for which communities were significantly different (Fig. 5b; Supplementary Table 1). Besides distance, both altitude and canopy openness, and to a lesser extent density of large trees played a major role in driving community composition of rare species (Fig. 9a, zeta order 2). Similar results were observed at zeta order 10, but TreM diversity, TreM density and volume of deadwood became more impactful, while canopy

openness had less of an effect. Curve slopes indicated a high but discontinuous and decreasing impact of canopy openness on community changes above ~0.2 (rescaled value), while community composition had a relatively constant sensitivity to altitude at low zeta orders overall (Fig. 9a, zeta order 2–20). Interestingly and according to the slope, community changes were sensitive to the density of large trees, quickly showing a slight unhook but with an important and continuous impact of this environmental driver at up to 0.2–0.4 (rescaled value) overall (Fig. 9a, zeta order 2, 10). Finally, the bigger the zeta order was, the more impact TreM diversity, TreM density and deadwood volume on-site had, as only these three environmental factors (aside from distance) were found to drive compositional turnover (Fig. 9a, zeta order 50). TreM at high zeta orders showed a similar impact trend as canopy openness at low zeta orders on community composition, plateauing at ~0.9 at zeta order 20, 1.5 at zeta order 28, 3 at zeta order 40 and 5 at zeta order 50, while deadwood amount followed patterns of the density of large trees (Fig. 9a, zeta order 20–50). Interestingly, TreMs density also affected overall compositional turnover but had less impact than TreM diversity in driving community composition. The nine tested variables explained 20–40% of the species composition turnover variance across all zeta orders. Distance excluded, the eight remaining variables only accounted for 15–20% of the variation, hence most of the variance remained unexplained (Fig. 9b). This suggests a complex multifactorial effect of environmental factors, but not random assemblies (Fig. 4c–d; Supplementary Fig. 4c–d), with many variables yet to be explored, in driving the insect community composition turnover of both rare and common species.

## Discussion

Marked declines in insect abundance, biomass and species richness have recently been quantified in Europe[2]. The causes of insect decline are multifactorial with rapid climate change identified as one of the major drivers. Here we investigated whether forest disturbances such as drought-induced forest decline and subsequent salvage logging have an impact on flying insects. Surprisingly, insect richness remained stable regardless of the extent of dieback or salvage logging. However, insect species composition changed

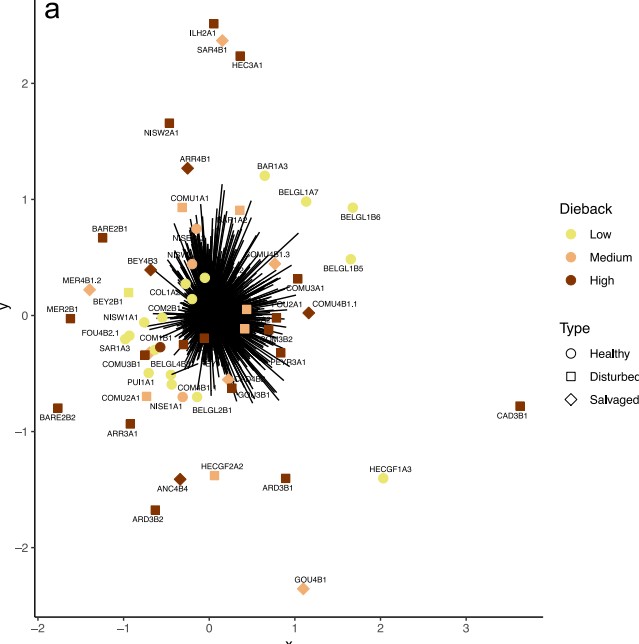

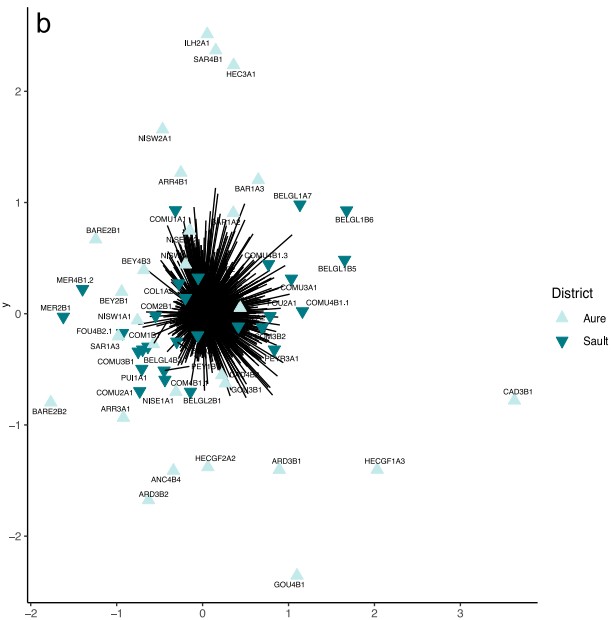

**Fig. 5 Variation in community composition across dieback gradient, stand types and geographical districts.** Gaussian copula ordination plots representing variations in insect community assemblages in regards to: **a** dieback level conditions (i.e. low, medium and high) and stand types (i.e. healthy, disturbed and salvaged), and **b** to the two different geographical districts sampled. Mvabund analyses of community dissimilarities were performed using 999 bootstraps. Each black line represents the resultant of both factors 1 and 2 of the traitGLM Gaussian copula ordination for each studied plot reduced to 95% of the 2.5 set alpha-ratio. Ordination **a** highlighted higher similarity in community composition between low dieback level plots than the other two dieback levels, more similar to each other but expressing high variability in community composition. No grouping of salvaged-logged plots in terms of community composition could be distinguished. Ordination **b** highlighted the greater dissimilarity in terms of community composition within plots of Aure valley compared with Sault plateau that hosted species assemblages more similar to each other.

significantly with the level of dieback, driven mainly by turnover among rare insect species. Comparable changes in insect species composition but stable insect species richness have been observed in response to pest-induced forest dieback of other tree species[22,33], and the fact that similar observations have been made in aquatic insect communities over time[17] suggests that our findings are indicative of a broad ecological pattern.

While species richness remained similar across the forest dieback gradient, community composition of the orders Lepidoptera, Hemiptera, Diptera and Hymenoptera changed significantly. Furthermore, all functional groups studied (i.e. floricolous/non-floricolous and parasitoids/non-parasitoids) also showed significant compositional turnover driven by forest dieback. In line with previous study, we demonstrate how widely impacted biodiversity can be from dieback regarding both taxonomic and functional assemblages[34]. Only Coleoptera had no detected effect of disturbances in general on the composition of their communities. This is surprising as Coleoptera have been shown to be significantly affected by both forest diebacks and salvage logging[35]. But while they were the sole group bearing significant change in species richness across districts, this absence of community change across dieback gradient had been already observed on saproxylic beetles in the same sampling area[36] but may also be an artefact of the use of Malaise traps which do not collect many Coleoptera and a biased or lower sampling efficiency of Coleoptera in Sault plateau.

In contrast to forest diebacks, salvage logging practices had no impact on species richness nor on community composition in our studied area of the Pyrenees. This result has to be taken with caution since the salvage logging intensity and the associated amount of deadwood removed from our plots might have been too small to significantly impact insect fauna. Because the slopes of the sampled montane forest plots were steep and logging mechanically performed using skidders and cables, we suppose that both reasoned management practices and the difficulty to harvest deadwood mitigated salvage logging impacts on insect biodiversity. In addition, winning and losing taxa from salvage logging[13] may have had overall counter-balancing effects. Indeed, we found a weak but expected negative effect of salvage logging on the prevalence of Coleoptera similar to a meta-analysis[13]. We think the first explanation of low salvage logging intensity is more plausible and supported by the fact that we did not detect any impact of salvage logging on deadwood amount either in our study sites[36]. Furthermore, as sampling occurred in non-old-growth forests, salvage logging may have left a sufficient amount of coarse woody debris on-site to prevent severe impacts on insect diversity. Insect communities in managed stands such as those we studied may be poorer, more homogenous and have fewer taxa associated with deadwood than in more mature forests. This may have reduced our ability to detect changes in species richness. In addition, our sampling was carried out more than 10 years after the onset of climate-induced tree dieback and subsequent logging, perhaps providing time for a recovery of species richness, even though extinction debt (i.e. a time-delayed negative response of a taxon to an environmental disturbance) was observed in Diptera after 29 years in *Quercus* spp. dominated forests of Southern France[37].

The lack of variation of species richness of insects with dieback level found in our study contrasts with other studies that show a positive effect of forest dieback on insect biodiversity[14,15]. However, those studies are based on the response of well-studied groups, i.e. "those insects that we love and cherish"[38]. Indeed, Moretti et al.[15] highlighted that insect diversity could respond both negatively and positively to fire-induced forest diebacks after examining the datasets at lower taxonomic level. They found that the winners were among the most studied groups of insects

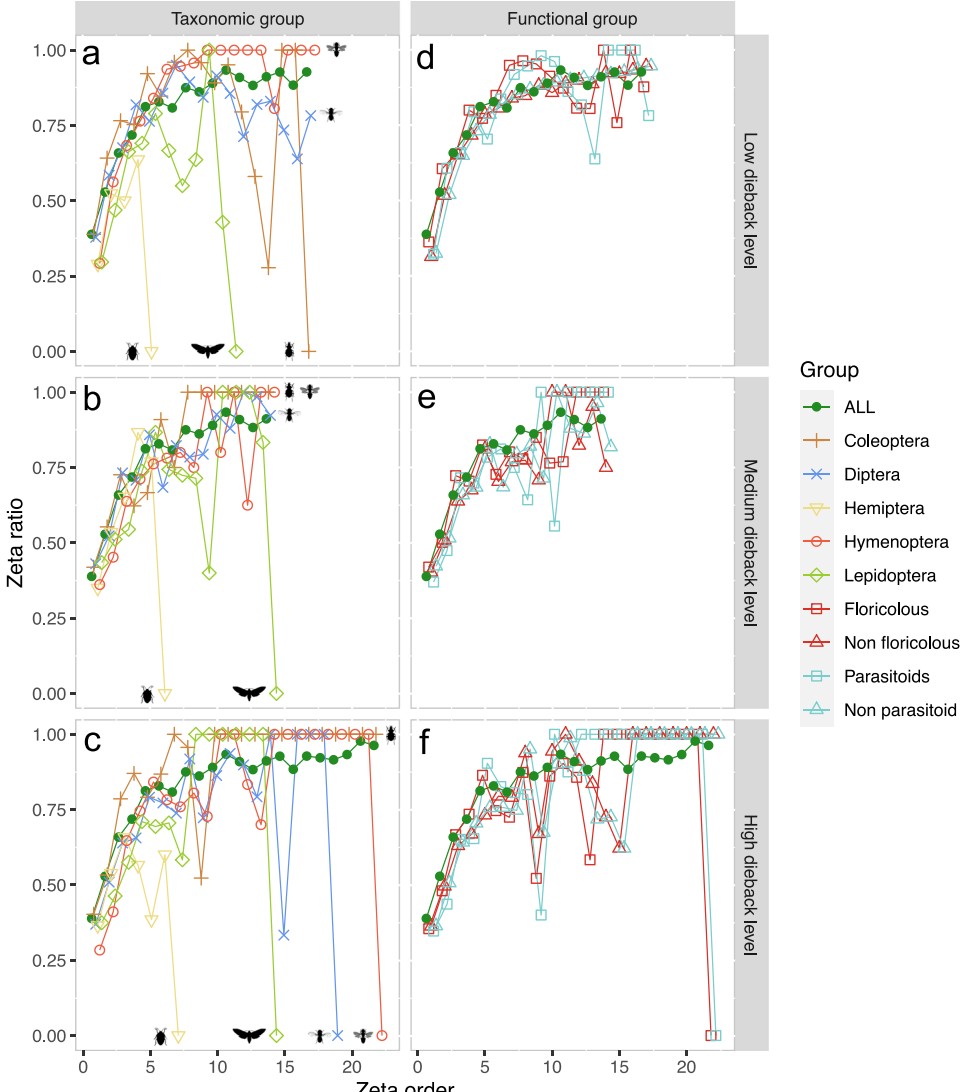

**Fig. 6 Effect of dieback on community composition for insect orders and functions.** Representation of the species retention rate (i.e. zeta ratio) per plot (i.e. zeta order) following nearest-neighbour plot combinations scheme (NN) with parameter sample set to 5000 and Monte–Carlo (mc) sampling for low, medium and high dieback levels, respectively (**a–c**) for the five main insect Orders (Coleoptera, Diptera, Hemiptera, Hymenoptera and Lepidoptera) and (**d–f**) for the four main ecological functions recovered from taxonomic assignment (floricolous/non-floricolous and parasitoid/non-parasitoid species). Green line with plain dots represents mean species retention rate of the total dataset in each respective dieback category. Increasing curves express that common MOTUs are more likely to be retained in additional samples than rare ones (with presence of common species over all plots if zeta ratio = 1) and decreasing curves indicate species turnover. In our study, no core of common species could be sampled for Coleoptera at low dieback level as well as for Lepidoptera and Hemiptera throughout the entire dataset for each dieback category, respectively. For both Diptera and Hymenoptera, common core of species was observed within all dieback level plots except between high dieback level ones. Similarly, drop and lack of structure in common species was detected for floricolous and parasitoid functional assemblages within high dieback level plots while stable for the other two dieback levels, with this functional turnover being driven by dipteran and hymenopteran species turnovers at high dieback.

(i.e. hoverflies, bees, social wasps and ground beetles) while only one losing insect group was identified (weevils)[15]. This implies that other, unidentified groups were potentially "losers" but less likely to be reported. Similar bias toward these well-known insect groups is also identifiable in studies on pest-induced forest dieback with an additional focus on red-listed species[10], or in the studies considered in Thom and Seidl's meta-analysis[14]. From this detailed taxonomic view[15] and recent comprehensive studies[22,33], an observation of a positive response of insects to disturbances must be taken cautiously as it could be biased towards the response of the better studied insect groups. Previously reported global patterns may thus not be fully representative. Here, by including hyper-diverse and understudied groups ("dark taxa")[32], the overall stable species richness that we found across both the

dieback gradient and stand types highlights a nuanced and idiosyncratic response of insect orders and functional groups such as floricolous/non-floricolous and parasitoid/non-parasitoid insects to forest disturbances. In addition, our results in both insect species richness and compositional turnover emphasize the limitation inherent to the sole use of species richness as a metric[39] and the need to look more closely at community composition and functional changes to detect the response of insects to disturbance[40]. Indeed, we show that various dieback levels would rather promote different insect community assemblages and be part of natural succession dynamics (i.e. ecological change occurring in a predictable way after disturbance), with equally weighted response of species either winning or losing throughout the dieback intensity gradient. Hence, management policies based

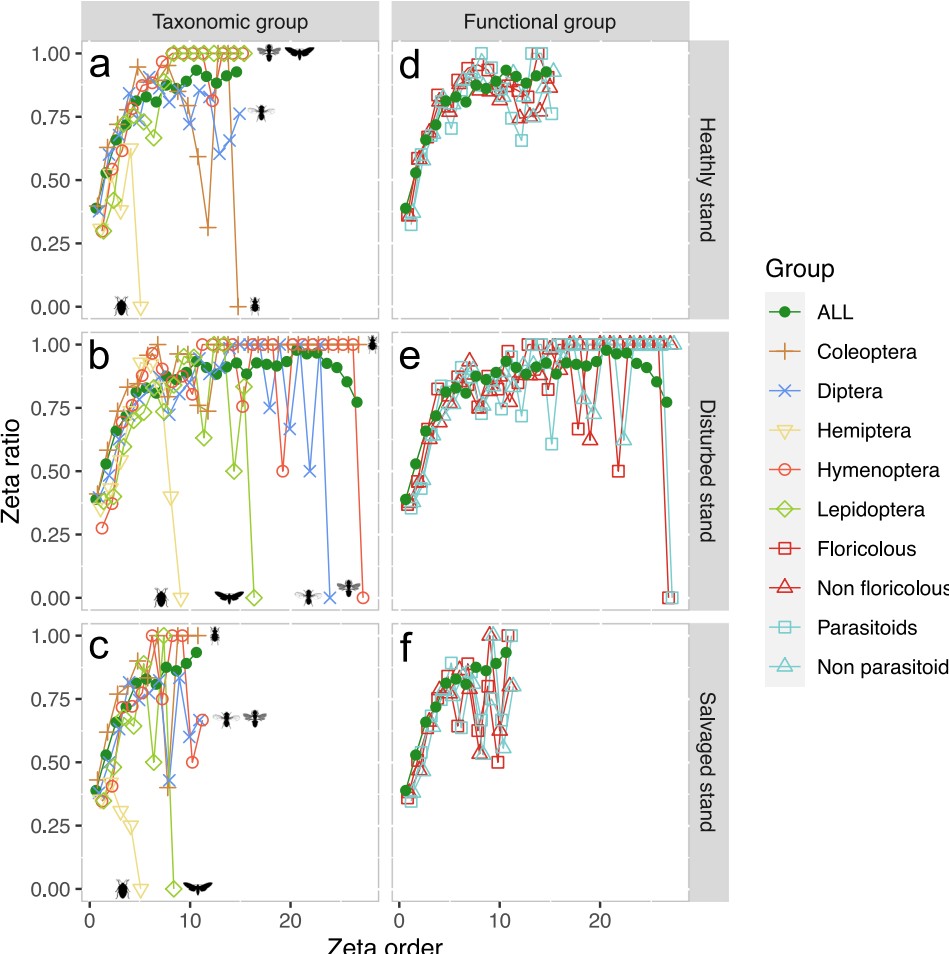

**Fig. 7 Effect of stand type on community composition for insect orders and functions.** Representation of the species retention rate (i.e. zeta ratio) per plot (i.e. zeta order) following nearest-neighbour plot combinations scheme (NN) with parameter sample set to 5000 and Monte–Carlo (mc) sampling for healthy, disturbed and salvaged stands, respectively, **a–c** for the five main insect Orders (Coleoptera, Diptera, Hemiptera, Hymenoptera and Lepidoptera) and **d–f** for the four main ecological functions recovered from taxonomic assignment (floricolous/non-floricolous and parasitoid/non-parasitoid species). Green line with plain dots represents mean species retention rate of the total dataset in each respective dieback category. Increasing curves express that common MOTUs are more likely to be retained in additional samples than rare ones (with the presence of common species over all plots if zeta ratio = 1) and decreasing curves indicate species turnover. In the present case, we found no core of common species for Hemiptera within each studied stand type. For clarity purposes, Hemiptera are hereafter not considered. Within healthy stands, only Coleoptera had no common species retained, while Diptera had a slight drop in the probability of common species occurring, but yet not a complete compositional turnover. Lepidoptera, Diptera and Hymenoptera were all impacted by disturbed stands with complete compositional turnovers within studied plots, and similar observation could be noticed for parasitoid and floricolous functional assemblages. Meanwhile, Coleoptera had a constant core of common species across the considered plots, and similar observation could be made within salvaged plots. Again, Lepidoptera assemblages were fully heterogenous across salvaged-logged plots, and both Diptera and Hymenoptera expressed a lower probability of sampling common species, but yet not showing a complete change in community composition.

only on species richness metrics and letting be forest diebacks to favour biodiversity may hold potential risks by driving a catastrophic decline of insects through a potential homogenization of dieback levels and associated species assemblages with more generalist species. Albeit Borges and collaborators[40] highlighted the issue in the very distinct island ecosystem, their concern can be extended to freshwater streams[17] and disturbed forest environments as in the present case.

Looking at community compositional changes in a more specific way, zeta decline fitting a power-law regression implied a compositional turnover of the total dataset driven by environmental factors rather than stochastic events[29]. The zeta ratio analyses showed that species are indeed retained differently within each dieback category. Thus, the lack of core set of species observed for Lepidoptera may be attributed to a relatively low moth diversity associated with silver fir-dominated forests with

community composition mainly shaped by turnover processes[41]. On the other hand, sampling efficiency could explain the absence of common species retained for Hemiptera for the different dieback categories or stand types tested[29]. Nevertheless, the prevalence (i.e. the presence probability of a taxa within all the plots grouped in a given environmental category) of these two orders remained positively correlated with dieback in general, in accordance with previous observation in which open canopy deriving from diebacks favours heliophilous and flower-visiting insects[15]. Interestingly, the prevalence of Coleoptera was favoured by low dieback level while species richness across the dieback gradient was similar and the species retention rate decreased. Even though we could not finely assess feeding guilds, these observations may support high turnover processes and competition deriving from specialist species' population decrease as factors shaping Coleoptera communities[9]. However, salvage logging

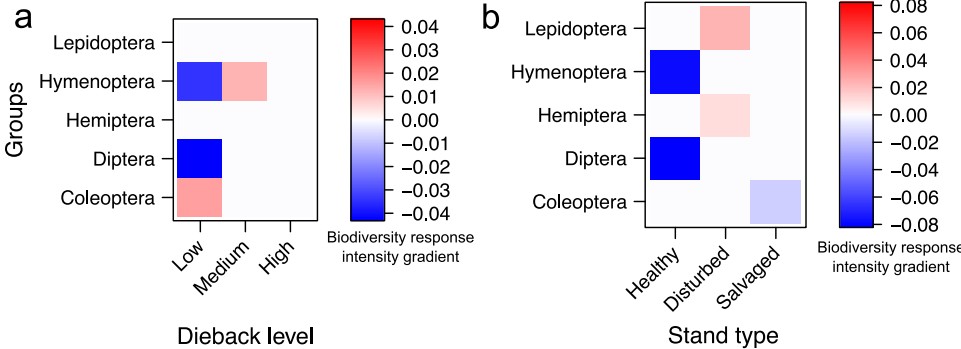

**Fig. 8 Insect orders winning and losing from environmental conditions.** Heatmap representation based on traitGLM analyses with LASSO penalty of insect orders' prevalence according to environmental conditions: **a** forest dieback level (i.e. low, medium and high) and **b** stand types (i.e. healthy, disturbed and salvaged). Negative and positive interactions and their intensity in terms of biodiversity response to environmental conditions for each of the five most represented orders are highlighted by a continuous colour gradient spanning from blue to red, respectively. We found specific responses for each insect orders' diversity in regards to dieback levels and stand types. For instance, the low prevalence of dipterans and hymenopterans observed at low dieback level while relatively high prevalence for coleopterans. However, hymenopteran prevalence was greater in forests of medium dieback level. Coleoptera was the only order with prevalence negatively impacted by salvage logging. Finally, disturbed plots showed a relatively high prevalence of both Lepidoptera and Hemiptera.

had a slight but significant negative impact on Coleoptera prevalence, in line with previous observations and consistent with the fact that a large proportion of forest-dwelling Coleoptera are saproxylic[13,36,42]. In addition, each dieback gradient hosted a specific core set of one or more common species composed of Diptera and Hymenoptera at both low and medium dieback levels. The reduced openness associated with healthy stands—hence low dieback level—might also support the reduced prevalence of both Diptera and Hymenoptera observed in these environments, as most of the species recovered for these orders are considered floricolous. Dieback might also increase environment complexity and support the positive effect of medium dieback level that we observed on Hymenoptera, especially parasitoid ones[23], rather than a negative bottom-up effect of the host condition on parasitoid species[24].

A compelling observation in terms of functional diversity is that while both Diptera and Hymenoptera displayed strong compositional turnover at high dieback level, their drastic changes in community composition drove in turn the turnover observed for floricolous and parasitoid functional assemblages. It is known that both floricolous and parasitoid Hymenoptera can indeed be significantly impacted by structural changes in silver fir-dominated montane forests[43]. However, our results emphasize the ecological importance of some species-rich yet often overlooked taxa on wide functional groups. Furthermore, it highlights the need for more work to investigate whether climate-induced forest dieback can deeply impact functional efficiency and ecosystem services through these taxa[3]. While these broad functions remain, the observed complete change in their respective species cohorts through the dieback gradient might impact trophic relationship at finer scales and species-dependent interactions.

In a more detailed view, within each studied order or functional group, the winner and loser taxa were relatively rarely occurring species in our dataset. This pattern is comparable to previously reported taxonomic dissimilarities driven by rare species[34] and consistent with models showing that selective perturbations directed towards rare species generate more dynamical effects on the ecosystem[44]. As rare species have a positive impact on ecosystem multifunctionality[45,46], these rare taxa have high patrimonial value and thus their importance in forest ecosystems functioning should be further investigated[34].

To investigate for environmental features in an attempt to explain these complex compositional changes, we first found that

geographic distance—regardless of the zeta order considered—played a significant role as a macroecological factor shaping the different insect communities of the two districts and, as for the aforementioned winner and loser taxa, mostly taxa with low prevalence shaped compositional specificities of each district. As expected, many other forest features tested (i.e. altitude, density of large trees, canopy openness, total volume of deadwood and both TreM diversity and density) also had a significant effect in shaping insect community composition, indicating a complex response to disturbances. Remarkably, environmental factors explaining most of the compositional variance differed according to the observed zeta ratio. At low scale (zeta order = 2), community changes were mainly driven by canopy openness, density of large trees and altitude. This result supports the role of distance and stand structure heterogeneity in favour of biodiversity[47] and is in line with previous observations on these features affecting the rare species composition at low zeta order[22]. In addition, slope trends for both altitude and canopy openness at low zeta orders reflect previous observation, with respectively continuous (full range) and discontinuous (between 0 and 20% of its range) impacts on community compositional changes[22]. However, these environmental features at large scale (zeta order >10) were no longer the main drivers of biodiversity changes. Instead, both TreM diversity and density, as well as the amount of deadwood were the main factors influencing community composition when considering the whole dataset. Interestingly, we highlight that TreM diversity, TreM density and amount of deadwood impacting ranges on community changes at high zeta orders have similar impacts as canopy openness and altitude at low zeta orders, respectively. This finding may result in various management strategies to promote biodiversity, with both canopy openness and TreM diversity manipulated in similarly limited way (i.e. around 20% increase maximum) in accordance with the size of the managed area, but without threshold in regards to deadwood amount on-site for wide areas. Here, we also highlight that TreM diversity had more impact than TreM density on insect diversity[48], similarly to deadwood diversity and amount in other studies[47,49]. However, as TreM diversity is partly linked to TreM density, it indirectly also has a great influence on biodiversity[48]. Furthermore, and contrary to previous report[47], we found deadwood amount to be as valuable as stand structure for biodiversity conservation, depending on the geographic scale managed[42], which further supports the

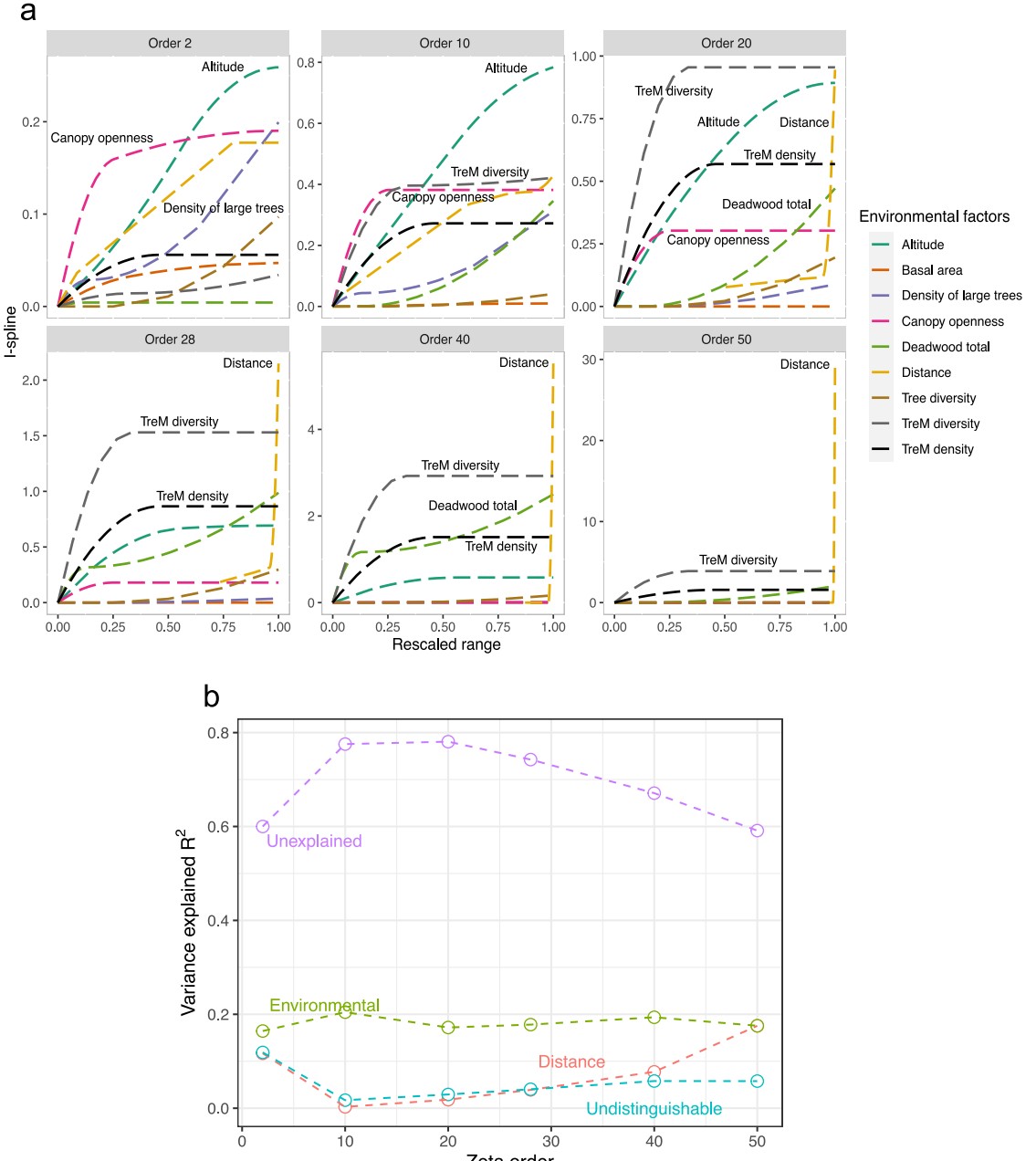

**Fig. 9 Environmental variables driving species community changes (zeta diversity changes). a** I-splines mean values of predictors from 30-rounds calculation with Sørensen-equivalent metric for seven environmental variables with their respective contributions to zeta diversity. Environmental contributions are shown for six different zeta orders (i.e. $\zeta_i$) here referring to plots compared together (from $\zeta 2$ to $\zeta 50$), with the main contributively environmental variables highlighted for each zeta order. We observed a shift in the key forest features explaining community changes according to geographic scale, with canopy openness and altitude as main environmental drivers at local scale ($\zeta 2$), replaced by tree-related microhabitats diversity and deadwood amount at large scales (from $\zeta 28$ to $\zeta 50$). **b** Proportion (in $r^2$) of the total zeta-diversity variance explained by environmental variables, distance, undistinguishable variables or unexplained, according to zeta order. The eight tested environmental variables accounted for 15–20% or the total variance in community changes regardless of zeta order, with distance explaining up to 20% of the total variance as well depending on the different zeta orders considered.

general call on acknowledging and considering the importance of deadwood to biodiversity[42]. Interestingly, a previous study successfully linked hoverfly diversity to plot connectivity and the combined effect of environmental factors from both stand and landscape scales[37]. The shift in environmental variables driving community assemblages according to the zeta order considered we highlighted here support such geographic scale effect and help disentangling the key drivers in action. This may potentially be extended to most aerial insects with higher dispersal ability,

which are the main representatives of our Malaise-trapped dataset. Hence, while species-specific response to environmental factors must be accounted for, geographic scale of the conservation area, especially through stand and landscape scales[37], as well as dieback intensity should also be considered when managing stand heterogeneity and environmental factors to favour forest insect biodiversity[47].

Finally, our study confirms DNA metabarcoding of Malaise-trapped samples to be an efficient approach for biomonitoring

changes in species-rich insect communities that include "dark taxa", which collectively represent the bulk of forest insect diversity[22]. While read-counts may not provide accurate abundance estimates for species, promising developments such as the spike-in method[50] may improve this capability. Nevertheless, our metabarcoding pipeline coupled with recent efforts to complete the DNA barcoding reference library of European insect fauna[19,51] allowed us to inventory nearly 3000 insect species, more than half of which were identified to species level, including hyper-diverse taxa such as Diptera and Hymenoptera[52]. These two taxonomic groups were the most diverse in our study, like in other Malaise trap environmental surveys[53,54]. Although Malaise-trapped samples are biased toward flying insects[55], the zeta decline analyses indicated that our dataset incorporated rare species as non-stochastic events and tracked changes in both species richness and community composition at both large and small spatial scales while overcoming sampling district discrepancies[29]. Our study is limited to a single year but provides a detailed account of insect diversity for the studied area and may serve for future monitoring of ecosystem recovery and biodiversity changes over time. Indeed, scaling up our approach to national and continental levels would help to monitor insect biodiversity and potential decline, to provide an understanding of the environmental drivers of biodiversity loss in a rapidly changing climate, and allow for regular assessment of the efficiency of conservation and forest management policies.

## Methods

**Study sites, stand description and insect sampling.** Fifty-six natural forest plots were selected for our study in 2017 in silver fir (*A. alba*) forests in the Central (Aure valley) and Eastern (Sault plateau) French Pyrenees (see Supplementary Data 3 for the complete plot list).

In each plot, tree dieback was assessed using the ARCHI method[56]. The 20 silver fir trees closest to the plot centre (defined by the Malaise trap), belonging to the dominant layer and with a diameter at breast height (dbh) >17.5 cm, were examined with binoculars. The count of dying trees and the proportion of affected trees (i.e. trees expressing signs of climate-induced dieback and comprising stressed + resilient trees) were used as a proxy of the decline at the stand level to define a three-category gradient: 23 "high dieback level" plots (at least 1 dying tree and >60% of affected trees), 15 "medium dieback level" plots (no dying trees and >60% of affected trees) and 18 "low dieback level" plots (no dying trees and ≥40% of healthy or resilient trees). Completely healthy stands are difficult to find in natural dieback context. Here, our low dieback level category plots also underwent the 2003 drought but recovered or were resilient, and therefore acted as healthier controls.

For each living or dead tree belonging to a fixed-angle plot (ratio 1/50), we recorded its status (living, snag, log), tree-species, dbh (minimum dbh to be recorded: 17.5 cm for living trees and logs, 7.5 for snags). Tree-related microhabitats (TreMs) were recorded on living trees and snags according to the Larrieu and collaborators' typology (47 types)[48]. TreM diversity corresponds to the number of TreM types recorded per plot while TreM density corresponds to the total number of TreMs recorded per plot. Very large trees were defined as living trees above 67.5 cm in dbh. Since we used the ratio 1/50 to fix the angle, basal area matches with the number of trees observed. Each deadwood item (length > 1 m) was measured in length and diameter to calculate its volume. Mean plot area was about 0.3 ha, depending on the dbh and location of the largest trees. In addition, canopy openness was evaluated by using a spherical densiometer[57] at the plot centre and 10 m from the centre in four crossed directions (thus 5 locations per plot). Canopy cover was calculated for each location as the proportion of the 68 points of the densiometer screen that was intersected by cover. Plot value resulted as the mean of the five counting.

Stand type was described on an area of 1 ha using the Index of Biodiversity Potential (IBP)[58]. The IBP is a biodiversity evaluation tool (according to Larsson[59]) combining ten historical, structural and compositional key factors for forest-dwelling species that are easily and directly measurable in the field[60]. The measurement of these factors provides a rapid assessment of the theoretical biodiversity hosting capacity of the stand and facilitate the identification of the key environmental variables manageable for better biodiversity potential. In our study, the main factors accounting for healthy, disturbed and salvaged conditions were stand structure (canopy openness), tree health (deadwood volume and number of living trees) and the awareness of management activities influencing deadwood volume and density of habitat-trees available on-site. As no plot was experimental, no controlled salvage logging activity was performed and deadwood volume was influenced solely by local forestry.

Insect sampling was conducted from late spring to early autumn in 2017 (May 15 to September 15) using 56 Townes-style Malaise traps with black walls and white roof[54]. One Malaise trap was placed at the centre of each of the 56 one-ha plots. Malaise trap sample-bottles were filled with a mixture of 20% mono-propylene glycol and 80% pure ethanol. Samples were retrieved once per month, resulting in a total of four samples per plot over the sampling period (124 trap days), for a total of 222 samples (two samples were lost). After collection, all samples were stored at 4 °C until laboratory processing.

**Laboratory processing and DNA extraction.** Insects were first filtered from the trapping solution and rinsed with ultrapure Milli-Q water to remove mono-propylene glycol residue. Insects were then placed within sterile and disposable Petri dishes on clean absorbing paper to dry overnight at ambient temperature.

Once dried, insects were size-sorted using decontaminated forceps. Insects larger than a European honey bee (*Apis mellifera* Linnaeus, 1758) were removed and only the head or a part of the abdomen was retained in order to reduce the biomass and improve the detection of rare or small species[61]. Insect bulk samples were then ground and homogenized into fine powder using disposable BMT-50-S-M gamma sterile tubes (IKA) with 10 steel beads with an Ultra Turrax Tube Drive grinder (IKA). Ground bulk samples were then conserved at −21 °C.

DNA extraction of 25 mg (±2 mg) of insect powder was performed on silica columns using a standard DNeasy® Blood & Tissue extraction kit (QIAGEN) (see: www.qiagen.com/handbooks for further information). For each ground sample, we took 25 mg (±2 mg) of insect powder in a 1.5-mL microcentrifuge tube using spatula previously decontaminated with 4% Decon® 90 solution and autoclaved. During powder sampling, empty 1.5-mL microcentrifuge tubes containing 200 µL ATL buffer were left open and changed every nine samples to control for potential cross-contamination with volatile insect powder and all were processed as extraction controls (EC). Mock communities of 248 Asian insect species were used as positive controls. Negative (NC), extraction (EC) and positive (PC) controls were processed down to sequencing. Lysis was performed with horizontal shaking overnight at 56 °C in 180 µL ATL buffer and 20 µL proteinase K. All vortex steps were replaced by handshaking to reduce DNA degradation and DNA was eluted into 80 µL of AE buffer following 15 min incubation on the silica column at ambient temperature, and a second elution with the previous eluate after 5 min incubation on the silica column. Each sample was quantified using Qubit® 2.0 fluorometer dsDNA High Sensitivity kit (Invitrogen) and eluate subsamples were diluted to 2 ng/µL. Both stock solutions and diluted eluates were stored at −21 °C.

A twin-tagging dual-indexing approach was used for the sequencing library preparation[62] (see Supplementary Data 4 for the complete primer list). We targeted a 313-base pair (bp) fragment of the mitochondrial DNA Cytochrome oxidase *c* subunit I (COI) gene. PCR amplification was done using the mlCOIintF forward primer 5′-GGWACWGGWTGAACWGTWTAYCCYCC-3′ and jgHCO2198 reverse primer 5′-TAIACYTCIGGRTGICCRAARAAYCA-3′[63,64]. To facilitate twin-tag multiplexing, each of the 96 primer pairs was synthesized with a 7-bp tag at their 5′ end, differing by at least three bp and avoiding G nor TT at the 3′ end to prevent a succession of the three same nucleotides once attached to the primers. Furthermore, A, T, CA, CG, GC and GT nucleotides were added to the tag to act as heterogeneity spacers[65] to increase complexity and shift the reading frame for better sequencing quality on Illumina MiSeq platform (Supplementary Data 4). Shifting was in accordance with green/red light balance on v2/v3 Illumina MiSeq technology[66]. Neither proof-reading nor hot-start Taq polymerases were used for PCR amplification due to Inosine bases in the reverse primer[67].

Prior to amplicon PCR, we tested the optimal number of cycles and the quantity of DNA added to the PCR-mix by quantitative PCR (qPCR) using a LightCycler® 96 Instrument (Roche) with the KAPA Library Quantification Kit (Illumina). The qPCR conditions were one preincubation step of 95 °C for 5 min, followed by 40 cycles of 95 °C for 30 s, 45 °C for 60 s and 72 °C for 90 s, with a final extension at 72 °C for 10 min and a high-resolution melting step of 95 °C for 60 s, 40 °C for 60 s, and an acquisition gradient from 65 to 97 °C. The qPCR reactions with a final volume of 15 µL were prepared with 7.5 µL of KAPA qPCR mix (2X), 0.3 µL of each forward and reverse primers (10 mM), 3.9 µL of extra pure molecular grade water and dilution series of 3 µL DNA template (set at different concentrations: 8, 4, 2, 0.4 and 0.2 ng/µL) were performed to investigate inhibitions. The optimal number of PCR cycles was designated by where the exponential phase transitioned to plateau phase. PCR amplifications were thus run for 23 cycles under the same conditions as the qPCR, in 25 µL reaction volume as follows: 2.5 µL of buffer green (10X), 0.5 µL of dNTPs (10 mM), 1 µL of each forward and reverse primers (10 mM), 0.2 µL of Metabion mi-Taq (5 U/µL), 16.8 µL of extra pure molecular grade water and 3 µL of DNA template at 2 ng/µL. Each sample was PCR-amplified three times in different 96-well plates with three different twin-tag sequences, to allow independent tracking of the three PCR products after amplification. PCR products were purified using CleanPCR magnetic beads (CleanNA) at a ratio of 0.8 µL per 1 µL of PCR product. Two steps of rinsing with 200 µL of 70% ethanol were performed before final elution in 25 µL TE buffer (1X). Purified PCR product was then quantified in duplicates on a FLUOstar OPTIMA microplate reader (BMG Labtech) with Quant-iT™ PicoGreen® dsDNA assay kit (Thermofisher) following manufacturer's protocol. Equimolar pooling of the samples was carried out for each plate with a total DNA quantity of 1 µg of purified amplicon for a final volume of 60 µL. A second indexing for all PCR plates was conducted using the

TruSeq DNA PCR-free Library Prep kit for PCR-free ligation (Illumina) with 18 different indices (one per 96-well plate) following the protocol given in Leray et al.[62]. Libraries consisting of the total 222 samples were sequenced on five MiSeq runs – v3 (Illumina) with 600 cycles.

**Use of positive controls and blanks in demultiplexing.** Mock communities of known species composition with DNA barcode sequences available were used as PCs. The 248 species used in the mock community all have distributions restricted to east Asia, and hence allow for the examination of tag-jumping in the Illumina sequencing process. PCs were sequenced and bioinformatically treated the same way as bulk samples to ensure comparability. We focused on a restrictive approach, favouring false negatives to false positives (see Alberdi et al.[68] for more information on bioinformatic set-up). To decide both on the number of PCR replicates in which MOTUs must appear and minimum read numbers per PCR, we used BLAST+[69] on the PCs to find the set up in which we could recover the most MOTUs with 100% match and no MOTUs below 97%. Both NC and EC were checked for reads after demultiplexing to ensure that there was no cross-contamination and removes potential MOTUs recovered in controls from the total dataset.

**Reads demultiplexing, taxonomic and functional group assignments.** AdapterRemoval ver. 2.2.2[70] was used to trim the twin-tagged adaptors of the reads and we employed sickle ver. 1.33[71] to perform paired-end quality trimming. Error correction using Bayes Hammer was done via SPAdes[72] ver. 3.12.0, followed by paired-end merging with PandaSeq[73] ver. 2.11. PCR replicates were treated following the DAMe pipeline[74]. Heatmaps of primer tag combinations versus read numbers were generated using R ver. 3.6.1[75] to control for indexing quality. Our twin-tagging approach allowed us to easily identify errors and problems at this step. Renkonen Similarity Indices (RSI value[76]) were calculated for unique sequences found in at least three PCR replicates and represented by at least three reads (see section 'Use of positive controls and blanks in demultiplexing' for arbitration of required numbers of PCR replicates and reads). Average read size was plotted in R to ensure filtering and trimming quality, and VSEARCH[77] ver. 2.8.1 was run to remove chimeras.

Sequences were clustered into Molecular Operational Taxonomic Units (MOTUs) with a 97% similarity threshold[78] using SUMACLUST ver. 1.0.0[79]. Clustering quality was controlled via R, using the LULU approach[80]. Taxonomic assignment was performed by comparing nucleotide sequences with the BOLD System database[27] in April 2019 using a 97% threshold similarity[22] with bold R-package ver. 0.9.0[81]. For each MOTU, only unambiguous taxonomic assignments were kept, meaning that when multiple names at a given taxonomic rank were attributed, only common and unique taxonomic assignments at higher taxonomic levels were kept. In the case of multiple MOTUs with an identical and unambiguous species name assignment, all related MOTUs and their respective presence in sampled plots merged into a single species-specific MOTU.

We first assigned each of the 258 recovered insect families to different trophic guilds (i.e. saproxylic, zoophagous…) based on the known ecological functions of both immature and adult stages. Then, each family was assigned or not to two main functional groups: floricolous and parasitoids. This was done based on the capability of a given MOTU to function either as a floricolous species and/or parasitoid (or none) during at least one part of its life cycle (therefore fulfilling the given function in the ecosystem overall) (see Supplementary Data 2 for final functional assignment of insect families). We identified the insect families with parasitoid and/or floricolous species based on the literature[82–84] and with the help of expert taxonomists.

**Statistics and reproducibility.** All statistical analyses were carried out in R ver. 3.6.1[75]. Read-sequencing data were transformed in incidence-based data to account for presence/absence of MOTUs given that abundance data from metabarcoding can be misleading for metazoans[85].

We generated accumulation curves using iNEXT package ver. 2.0.20[86] with incidence-based frequency dataset parameters and Hill number $q$ value set to 0 to reflect truly observed MOTU diversity, regarded as species proxy. Total number of MOTUs was plotted against both sampling units and sample coverage for the late spring–early autumn sampling period as well as for each 4-weeks time series (named May, June, July and August from their respective starting months). MOTUs belonging to the five most diverse taxonomic orders represented in our dataset (i.e. Coleoptera, Diptera, Hemiptera, Hymenoptera and Lepidoptera) as well as for the remaining 10 Orders (i.e. Blattodea, Ephemeroptera, Mecoptera, Neuroptera, Orthoptera, Plecoptera, Psocodea, Raphidioptera, Thysanoptera and Trichoptera) pooled into an "Others" category, were also plotted against sampling units to account for respective recovery success. We also used both Chao and first-order Jackknife (Jack1) methods to have different biodiversity estimates from the 'specpool' function of R package vegan ver. 2.5-6[87]. These estimators also extrapolate species richness within the sampled area in regards to sampling effort and based on incidence data.

Impacts of tree diebacks on species richness were assessed using generalised linear model (GLM). For the total dataset, each taxonomic order category of the five main representatives (i.e. Coleoptera, Diptera, Hemiptera, Hymenoptera and Lepidoptera) or over the four ecological groups (i.e. floricolous/non-floricolous

adults and parasitoid/non-parasitoid larval feeding guild), GLM was first tested with both Poisson and quasi-Poisson distribution, the latter always retained for its better fit. Then, each model tested the effect of the dieback gradient, stand types, their interaction as well as the effect of districts. When no interaction was found, a simpler additive model was tested. In all nine studied cases, quasi-Poisson distribution and additive model characteristics were retained, as follow:

(1)  model <- glm(richness ~ dieback + salvage + district, family = quasipoisson).
  We finally applied type II Anova from car package ver. 3.0-8[88] to test the significance of each parameter on the species richness of the studied category.
  To compare insect community composition across disturbances gradients (i.e. tree dieback intensity and stand types) and districts, we performed GLM on a generated mvabund object using mvabund ver. 4.3.1[89] as follows:

(2)  Mod_Tot <- manyglm(Tot_mvabund ~ X$Dieback + X$Type + X$District, family = "binomial").

Reported values are the two extremes resulting from a 10 runs range. Post hoc Holm correction was then applied on the 10 p-values recovered to assess for the significance of potential compositional changes in community compositions. Ordinations for these GLM were performed using ecoCopula ver. 1.0.1[90].

Prevalence of the five most represented taxonomic orders in regards to dieback levels and stand types was assessed by a "fourth-corner model" analysis using 'traitglm' function from mvabund ver. 4.3.1[89] R package. For each insect MOTU, taxonomic order was defined as a trait and GLM analyses run to define traits significantly associated to environmental covariates.

To determine potential indicator species for each dieback gradient, we used IndVal analysis from 'multipatt' function of indicspecies R package ver. 1.7.9[91]. Again, reported p-value ranges are resulting from 10 independent IndVal runs with post hoc Holm corrections subsequently applied.

We also assessed the importance of rare species—definable from multiple dimensions (i.e. low abundance, habitat specificity, restricted range size, unfrequently encountered, etc) of rarity[92]—in species assemblages specific to both the gradient of dieback intensities and the sampling districts. Here, we specifically defined a rare species only from its frequency of appearance in a given dieback category or district, with occurrence (i.e. number of independent plots of each relative category in which we found the species) as an index of relative abundance. Indeed, neither temporal rarity nor true abundance or read-based abundance (as counts of individuals or of sequencing reads) could be used with our study design with non-independent monthly replicates and metabarcoding approach. Hence, MOTUs more frequently sampled throughout the plots were defined as more abundant and more common. We therefore generated heatmaps of relative abundance using the phyloseq ver. 1.30.0[93] package for R. As rare species can thus be virtually locally abundant in a given dieback category or district by the use of occurrence only to define each MOTU rarity, relative abundance used in heatmaps was generated from 100 randomized bootstraps of the total dataset to virtually increase sampling effort. Bootstrapped plots were then concatenated to recover a bootstrapped score giving the relative abundance. From this score, we could infer rarity in terms of habitat specificity, with common species occurring across spatial scale and/or dieback gradient, and rare species occurring only in a few plots in a particular district or level of dieback. Each of the three dieback categories (i.e. low, medium and high dieback levels) was rescaled to account for the difference in sample representativeness, as such that each category was depicting one-third of the total dataset. Each MOTU relative abundance was then calculated in percentage from the bootstrapped score for these rescaled categories, so that a MOTU present in all plots of one particular dieback category was present at 33% in the respective category, for a total of 100% across the total dataset if appearing everywhere. Regarding abundance per district (i.e. Aure valley and Sault plateau, as well as total dataset), MOTUs' relative abundances were calculated in percentage based on the bootstrapped score with no further rescaling per district.

Species compositional changes along the total sampling area were assessed by calculating the number of species shared across all 56 experimental plots using zeta diversity (i.e. extension of beta diversity over '$i$' zeta order ($\zeta_i$) or plots)[28] using zetadiv R package ver. 1.2.0[30]. Two models were computed to perform plot comparisons either following a nearest-neighbour (NN) scheme or considering every comparison possible (ALL scheme) to assess for heterogeneity driven by distance. For all zeta-diversity analyses, sample number for site combination was set to 5000, with Monte–Carlo sampling performed for NN and ALL models. The two models (NN and ALL) were also fitted both to power-law (pl) and exponential (exp) regressions to define whether the signal in our dataset was driven more by ecological niche and rare species turnover rather than by stochasticity, respectively[29]. To assess the better fit to a model between pl and exp, we chose the lowest score from Akaike Information Criterion (AIC), which imposes penalty to each model in regards to its number of parameters, and thus favour models satisfying parsimony criterion[94]. For both dieback gradient (i.e. low, medium and high dieback categories) and stand types (i.e. healthy, disturbed and salvaged), zeta ratio (i.e. the species retention rate, giving the probability of a common species to be retained in the community across zeta order here referring to plots) was generated for each of the five most represented taxonomic orders and the four different functional groups identified, as well as for the mean zeta ratio of the respective dieback category or stand type on the total dataset. Zeta ratio

calculations were performed with all sites kept for each category, including those without focal taxa.

Estimations of environmental variables contribution to insect species assemblage changes were performed with zeta multi-site generalized dissimilarity modelling (zeta.msgdm) in zetadiv R package ver. 1.2.0[30] at different zeta order, ranging from 2 to 50. As for zeta-diversity analyses, zeta.msgdm model was run with the number of samples fixed to 5000. After checking for non-collinearity, a total of nine environmental variables were tested with zeta.msgdm: geographic distance to the nearest plot, altitude, canopy openness, total amount of deadwood in $m^3$ per ha, the basal area per ha, the tree diversity per ha, the density of very large trees ($\emptyset > 67.5$ cm) per ha and both the TreM diversity and density per ha. zeta.msgdm was calculated using I-spline models[95]. Zeta.msgdm model was performed using Sørensen-equivalent metric and run over 30 rounds to obtain stable I-pline response curves of those predictors.

**Reporting summary**. Further information on research design is available in the Nature Research Reporting Summary linked to this article.

## Data availability

All datasets used for analyses are publicly available on Zenodo[96] (https://doi.org/10.5281/zenodo.5653307) or at the following GitHub repository: https://github.com/Lucasire/Malaise_FR_2017. Supplementary Data are publicly available on Figshare[97] at the following https://doi.org/10.6084/m9.figshare.16975636.v1. Raw sequencing data are available on NCBI at the following accession number: PRJNA702908.

## Code availability

All scripts used for bioinformatic demultiplexing and analyses are publicly available on Zenodo[96] (https://doi.org/10.5281/zenodo.5653307) or at the following GitHub repository: https://github.com/Lucasire/Malaise_FR_2017.

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

## Acknowledgements

We are grateful to all forest stakeholders—owners and managers—that allowed us to perform fieldwork in their stands. We are thankful to Laurent Burnel, Jérôme Molina, Sylvie Ladet, Carl Moliard, Jérôme Willm, Wilfried Heintz, Denis Sabadie and Benoit Nusillard for the fieldwork, field and GIS support, as well as to Kévin Speder for his help in the molecular lab. We are also thankful to Yohann Graux for his help with QGIS mapping and Rodolphe Rougerie for his valuable comments on the manuscript. Library preparation for sequencing as well as sequencing on Illumina MiSeq platform was performed at the Berlin Center for Genomics in Biodiversity Research (BeGenDiv). This research is part of the international project CLIMTREE "Ecological and Socio-economic Impacts of Climate-Induced Tree Dieback in Highland Forests" within the Belmont Forum Call: "Mountains as Sentinels of Change". The French team (L.S., A.B., B.C., J.C., C.B., L.L., E.A.H. and C.L.V.) was funded by the French National Research Agency (ANR) (ANR-15-MASC-002-01). L.S., A.B., E.A.H. and C.L.V. were also supported by FEDER InfoBioS (EX011185). L.S was partially supported by the German Academic Exchange Service (DAAD) (Short-Term Grant 57440917). The German team (P.S.Y., S.T., J.M. and M.T.M.) was funded by the Deutsche For-schungsgemeinschaft (DFG) (MA 7249/1-1) and the Federal Ministry of Education and Research (BMBF) (Förderkennzeichen 033W034A). W.C. and D.W.Y. were supported by the Strategic Priority Research Program of the Chinese Academy of Sciences (XDA20050202), the National Natural Science Foundation of China (41661144002—CLIMTREE grant—, 31670536, 31400470, 31500305), the Key Research Program of Frontier Sciences, CAS (QYZDY-SSW-SMC024), the Bureau of International Coop-eration (GJHZ1754), the Ministry of Science and Technology of China (2012FY110800), the State Key Laboratory of Genetic Resources and Evolution

(GREKF18-04) at the Kunming Institute of Zoology, the University of East Anglia, and the University of Chinese Academy of Sciences. D.F. was funded by the Italian CNR—Dipartimento Scienze del Sistema Terra e Tecnologie per l'Ambiente (CNR-DTA) (DTA.AD001.027VB-V). Credits for the insect shapes under CC BY 3.0 used: M. Broussard; G. Monger; M.T. Keesey, T. Assmann, J. Buse, C. Drees, A.-L.-L. Friedman, T. Levanony, A. Matern, A. Timm, & D.W. Wrase.

## Author contributions

This study was conceptualized and designed by C.L.V., E.A.H. and C.B. Forest plot selection and sampling were designed by L.L. and C.B. Sample processing, wet-lab experiments and sequencing were performed by L.S., P.S.Y. with the help of A.B., B.C. and M.T.M. Bioinformatic analyses were done by L.S. and P.S.Y. with the help of D.W.Y. and C.W. Ecological and environmental analyses were conducted by L.S. and C.W., with the help of J.C., C.B., D.W.Y. and D.F. L.S. led the writing of the manuscript. All authors contributed substantially to the interpretation and discussion of the results, with M.T.M., D.W.Y., S.T., J.M., E.A.H. and C.L.V. contributing substantially to the revision of the article. All authors approved the submitted version.

## Competing interests

The authors declare no competing interests.
