## [Peer Review File · Communications Biology]

Reviewers' comments:

Reviewer #1 (Remarks to the Author):

The authors studied the insect diversity in montane Pyrenean forests dominated by silver fir (i.e., sensitive to drought) because the effects of dieback (heatwave in 2003) and current salvage logging on associated insect fauna is unknown. In 2017, the authors collected 224 samples from Malaise traps (monthly), DNA metabarcoded the samples using 313bp of COI, and ecological functions were attributed using family-level information. They analyzed the data using generalized linear models to determine the dieback levels and salvage logging influence the structure and diversity of insect communities and functional guilds. They further analyzed the data using zeta-analyses in multi-site generalized dissimilarity modelling and fourth-corner model.

Positive feedback: The authors performed sophisticated analyses to study the influence of dieback and salvage logging on associated insects. The authors justified the bias of Malaise traps in sampling Coleoptera.

My comments/recommendations:

1) It would be great if the authors can move the Supplementary Fig. 1 "Map of the study area with sampled plot description" to the main body of the manuscript. It would be easier for readers to read/follow.

2) Overall, the flow of the discussion is not smooth. "Flow and connectivity allow readers to follow the thread of the argument from one sentence to the next and from one paragraph to the next." <https://www.sheffield.ac.uk/ssid/301/study-skills/writing/academic-writing/paragraph-flow-connectivity>

The flow of the first two paragraphs is OK; the last sentence of the first paragraph is connected to the first sentence of the second paragraph. However, the flow of the third paragraph to the fourth paragraph (and also other paragraphs) is not good. The third paragraph discusses forest dieback and the fourth paragraph discusses salvage logging. There is no connection between the last sentence of the third paragraph and the first sentence of the fourth paragraph. Please, improve the flow and connectivity of each paragraph.

3) Regarding the sequencing dept, please, include the total number of raw reads and the average number of reads with standard deviation (for example: 224,650 + 25,216 reads per sample) in the results.

4) Did you use proportion for the analyses? If yes, did you perform any transformation? How did you normalize the data? Which transformation did you use? Logit, arcsine, Freeman-Tukey double arcsine transformations or generalized linear mixed models? If you did not transform the proportion, please, justify.

5) Please, upload the resulting metabarcodes (final Operational Taxonomic Units; OTU) to the Barcode of Life Datasystems (BOLD).

6) Are there any endangered insects in the study area (montane Pyrenean forest)? If yes, do drought-induced forest dieback and subsequent salvage logging have an impact on the endangered insects?

Other minor comments:

7) Line 45, consider adding "the" between "explain change".

8) Line 178, consider changing "influences" to "influence".

9) Line 315, please, explain the artefact of the use of Malaise traps. For example (i.e., sample mainly flying insects).

10) Line 978, Itallize the species name "Aphidius ervi"

11) Line 911, 939, 966, 969, 991, 997, 1061, 1065, 1071, 1074, 1085, 1096, 1129, 1158 and 1178. Other than the first word with a capital letter, other words should be in small letters. For example Line 911, change "How Many Species of Insects and Other Terrestrial Arthropods Are There on Earth" to "How many species of insects and other terrestrial arthropods are there on Earth? ".

12) Remove the additional space on Line 370, 443, 544, 586 and 613.

13) Line 326. There is only one study cited here but the authors wrote "studies". Either change the word "studies" to "study" or cite more articles.

Reviewer #2 (Remarks to the Author):

This paper uses meta-barcoding data from Malaise trap samples to explore insect diversity patterns across two "disturbance" gradients (intensities of drought-induced dieback and salvage logging) in Pyrenean forests. The paper is well written and utilizes pretty complex modes of statistical data analysis (admittedly, these partly transgress what I am able to follow as an insect ecologist). The main finding is that species richness did not vary substantially across sites in the two gradients, but species composition did. This finding is not that surprising: differentiation diversity is usually more closely affiliated with environmental change than species richness. Multiple studies on selected target groups of insects have uncovered such patterns, but clearly the "all insect taxa" approach taken here has some novelty.

Moreover, the authors claim that observed compositional differences are largely due to 'rare' species, and observed patterns also vary across insect orders and/or guilds or functional types. Overall, the study appears well designed and thoroughly conducted. Nevertheless, I have a few rather general caveats that I feel should be addressed when revising the paper.

First, the paper is strictly based on incidence (rather than abundance) data. While there is little scope of doing differently with meta-barcoding data, it should at least be discussed (and preferably: explored!) how robust the results are in relation to the massive sampling problems that can arise from using incidence data with arthropods. For example, the number of species in common between samples, which is central to 'zeta diversity analysis', obviously can strongly depend on the completeness (and size) of samples. 'True' co-occurrences are notoriously under-rated with incidence data, because 'unseen' species are not necessarily 'absent'. See the many developments by Rob Colwell, Anne Chao and colleagues (like the Chao-Soerensen or Chao-Jaccard similarity measures, or also the NESS/CNESS measures), over the past 20-30 years in dealing with the many singletons and doubletons in biodiversity inventories of species-rich arthropods! At the very least, the readers should learn about sample sizes. Trap catches of arthropods may easily vary by 1-2 orders of magnitude between samples. Hence, it should be reported how large samples were, and whether sample sizes (e.g. dry mass of bulk samples?) systematically co-varied with forest characters or intensity of salvage logging. From your Figure 2 it is obvious that for all relevant insect orders, even at the landscape level, species richness was NOT exhaustively covered (the species accumulation curves are all still increasing well). Hence, at the level of individual samples coverage will be even MUCH lower, and this does affect your 'zeta estimates' (or in other words: you inflate/exaggerate 'rarity' just due to species missing/unseen in samples, viz. due to false negatives).

Second, I have problems with the mixed usage of the concept of 'rarity' in the paper. A species can be 'rare' in space, in time, and/or by numbers. Your analytical approach mixes that all up, and does not allow disentangling these very different dimensions of 'rarity'. Concerning numbers – you effectively have NO DATA on that using meta-barcoding, since you do not have counts or abundances of individual species. Honestly and openly say that, please. Rarity by time could and SHOULD be eliminated from your analysis, because the way you have subjected your data to 'zeta-analysis' exaggerates 'rarity'. Why? Because you used individual trap samples from the same sites, but taken at different times of the year, as putatively 'independent' data points. Which they are NOT! To illustrate my point: when you compare the numbers of shared species between samples, across space AND time (as you did) you will inflate the apparent 'rarity' due to simple (and massive!) phenological / seasonal species turnover in every temperate-zone ecosystem. For example, cockchafers (*Melolontha*) will only be there in May, so these seasonally very restricted

insects (as far as their adult volant stages are concerned) will exaggerate 'rarity' because obviously even at the very same place these beetles will not be found in any Malaise trap samples earlier or later at the same site. You could replace *Melolontha* with any other of the very many insects whose winged adults have a short emergence period, often of less than a month per site. However, this problem can easily be solved by aggregating repeated samples from the same site into ONE species list per SITE, and then re-do the analysis. Then, what would remain is one dimension of rarity, i.e. rarity by space (= by ecosystem or habitat), and this is the rarity you are talking about in all your paper. So, better configure the analysis such that it matches your hypotheses and research questions.

Third, it was unclear to me how exactly you assigned MOTUs to functional categories. Was that based on the ecological functions of the larval or adult stages? Was an ichneumonid wasp classified as a parasitoid (based on larval resource use) or as a nectarivore (based on adult feeding habits)? This applies to most holometabolous insects, notably Diptera, Hymenoptera and Lepidoptera, where the larvae and adults often occupy entirely different feeding niches. It must be explicitly stated whether your functional classifications refer to larvae or adults. Also you noted that you used insect families as proxy for scoring functions. I really disagree with that and use one simple example to illustrate my point, the Lepidopteran family *Erebidae*. Larvae of these may either be herbivores, detritivores, or fungivores. Adults may be nectarivores, may visit rotting fruits (*Catocalinae*), or may not feed at all (*Lymantriinae*, most European *Arctiini*). How, then, can you use family-level assortments for further functional analysis? Similar problems show up in many other more speciose insect families, e.g. *Syrphidae* in the Diptera, *Formicidae* in the Hymenoptera, etc.

Fourth, I would be eager to know how you have dealt with the many inconsistencies that one often encounters when working with the BOLD barcoding data base. The same insect species may show up there under different names, or there often is a mismatch found between taxonomic 'species affiliations' and so-called BINs in BOLD. Did you really not encounter any such inconsistencies when allocating your samples into 'species'? You obviously also, from understandable pragmatic reasons, used simplified thresholds of sequence similarity to delineate species, genera, etc. Many studies on barcoding with animals have, however, shown that a universal "barcoding gap" does not exist. Hence, it would be important to use your own large multi-species data to support whether, for example, the 97% similarity threshold (i.e. 3% divergence) at the species level is really valid, across all insects sampled in your study. Or whether, perhaps, a higher threshold might be more meaningful for insect order A, and a lower one for insect order B. Would your results change substantially if you employ, for example, a 2% or a 5% divergence threshold at the species level? For the sake of open science, I also strongly recommend to provide a full species x site matrix in an electronic appendix. Without such a "species list" nobody will ever be in the position to cross-check your results – yet repeatability and reproducibility are essential dimensions of science. These more general points all have the potential to alter the main results and conclusions presented in your paper.

I also came across a few minor, specific points.

L 69: too hot summer temperatures can also be harmful for insects ('heat stress')!

L 130: how reliable is family-based classification; especially in the light of your own wording above in the manuscript where you state that species-specific responses to environmental change may predominate? See also my doubts above on the validity of family-level generalizations.

Fig. 4: please say explicitly which ordination method you have used to create these diagrams. If this was an eigenvalue method: please report the eigenvalues (or explained variance, or fraction of trace of the relevant matrices, or whatever) at the coordinate axes.

Fig. 7: what precisely is the test statistics that you report here to take values between -0.08 and +0.08? I did not get that, perhaps I overlooked it – yet this must be clearly reported in the figure legend (which should be self-explanatory). Are these (standardized?) correlation coefficients? If so, the pertinent effect sizes would appear to be minute to me.

Reviewer #3 (Remarks to the Author):

This ms reports about the diversity of aerial insect fauna sampled by Malaise traps in a large number of *Abies alba* stands in the Pyrenees mountains throughout the summer 2017. The material was then barcoded and taxonomic units derived.

The amount of work is impressive as well as the data analysis.

I do not feel comfortable with the experimental design and hypotheses to test because:

- there are no control stands, i.e. stands where the tree decline associated with the drought event of 2003 was not present,
- the hypotheses are not clearly formulated or added at later time, i.e. when the results were obtained,
- the time span is limited to one year of observations

Reviewer #4 (Remarks to the Author):

Sire and colleagues conducted an impressive study on the effect of forest dieback and salvage logging on flying insects in silver fir forests. They used DNA metabarcoding to compile a large species by site matrix with 2972 MOTUs in 56 sample plots (suppl Table 1) and conducted some ordination analyses and zeta diversity analyses. The text is easy to read with a sound logic flow. I consider this a solid contribution to the research field.

I was asked to specifically check the soundness of the statistical modelling and analyses in this work. I don't have any major comments but only a few minor suggestions for the authors to consider so to further improve the clarity of their work.

L154: here, you mentioned 224 (=56 plots x 4 months) Malaise trap samples. However, in Suppl Table 1 you only reported 56 sample plots. I would suggest that you release the occurrence matrix for the 2972 MOTUs in 224 traps. This will allow readers to repeat your analyses. I also consider the complete dataset itself a valuable asset to the research field.

L155, 'iNEXT extrapolation estimation...': I understand that you mentioned the specific method used for richness estimation in the Methods section. It is still necessary to briefly mention here the method you used in iNEXT R package.

L167-176: You are discussing the patterns of distance decay of similarity, as well as the similarity decay with time elapses. As an option, you might consider to present the level of similarity as a function of geographic distance and time elapse.

L178: Perhaps clarify from the section you started using pooled monthly data for only 56 plots.

For the zeta diversity calculation and analyses, I presume that you used Monte Carlo sampling ("mc") instead of Exact method ('ex') due to the extremely large numbers of possible combinations for 56 plots. Please specify the number of site combinations used for calculating zeta diversity and also the MS-GDM.

Fig.4: The meanings of these clustered black lines in the centre of the plot should be provided in the legend.

In the zeta ratio plots, I noticed that there are a few retention curves have reached zero at high zeta orders, while others didn't (e.g. Fig.5). Please specify whether you have removed sites without the focal taxa in the calculation. I'm not suggesting that whether this practice is correct or not. Instead, they lead to different interpretations of the results (only slightly). So, it's better that you can specify it in the text.

L269: This section discusses the I-spline response curves of different predictors. The figures are insightful and contain quite a large amount of information. Besides the magnitude of a curve reflecting the importance (contribution) of a predictor, the specific shape (especially the slope at specific values of a predictor) also include detailed information: a greater slope is indicative of higher sensitivity of zeta compositional turnover to changes of the predictor at a specific value. For instance, compositional turnover is highly sensitive when canopy openness is less than 0.2

(rescaled value) but not so much when canopy openness is high (>0.2). I hope that the authors can look into such details in Fig.8 and add such details for each key predictors.

For the MS-GDM, it is also important to point out which metrics were used. I suspect that the authors used the raw zeta diversity. There are also different zeta metrics to choose (e.g. normalised, standardised, Sorensen-equivalent, Simpson-equivalent), with each providing slightly different meanings. In addition, if the model used a specific number of combinations for each zeta order, I would prefer that you run the model multiple rounds to check whether the I-spline response curves of predictors are stable or not. You can have a look of Latombe et al. (2019) *Journal of Biogeography* 46(10): 2299-2310.

Overall, I think this is a nice contribution to the research field. The authors have correctly implemented the analyses but with some further clarifications needed for transparency. Interpretations of the results can focus more on some interesting details.

Communications Biology submission 21-1178-T

Authors' response to reviewers

Summary

Global comments	2
Referee #1:	3
Referee #2:	5
Referee #3:	13
Referee #4:	14
References	20

Global comments

We have made the following changes to our manuscript in order to improve the clarity, although there were not specifically requested by the referees:

- Figures 1 & 5 colours have been modified, especially to be colour-blind friendly.
- The title has been changed to “Climate-induced forest dieback drives compositional changes in insect communities that are more pronounced for rare species”.
- Table I values of Chi-squares and *P*-values have been homogenized.
- A new citation (Ji *et al.* 2020, *Mol. Ecol Resour.*) has been added to the main text as an example of recently developed quantitative DNA metabarcoding approaches.
- Reference 23 “Péré *et al.* 2013, *Biol. Lett.*” has been changed for “Roland & Taylor, 1997, *Nature*”.
- Supplementary information on Family functional traits have been modified by removing trophic guilds and moved as Supplementary Table II. All Supplementary Table names have been modified throughout the text accordingly.

Referee #1:

1) It would be great if the authors can move the Supplementary Fig. 1 "Map of the study area with sampled plot description" to the main body of the manuscript. It would be easier for readers to read/follow.

RESPONSE: Supplementary Fig. 1 has now been moved to Fig. 1 in the manuscript. The text was modified accordingly and other figures were re-numbered.

2) Overall, the flow of the discussion is not smooth. "Flow and connectivity allow readers to follow the thread of the argument from one sentence to the next and from one paragraph to the next."

The flow of the first two paragraphs is OK; the last sentence of the first paragraph is connected to the first sentence of the second paragraph. However, the flow of the third paragraph to the fourth paragraph (and also other paragraphs) is not good. The third paragraph discusses forest dieback and the fourth paragraph discusses salvage logging. There is no connection between the last sentence of the third paragraph and the first sentence of the fourth paragraph. Please, improve the flow and connectivity of each paragraph.

RESPONSE: We have now rewritten the first sentence of the third paragraph and following (L. 342; 364-367; 395; 440-441) to improve the flow as requested.

3) Regarding the sequencing dept, please, include the total number of raw reads and the average number of reads with standard deviation (for example: 224,650 + 25,216 reads per sample) in the results.

RESPONSE: We have added the requested information in the result section L. 159-164.

4) Did you use proportion for the analyses? If yes, did you perform any transformation? How did you normalize the data? Which transformation did you use? Logit, arcsine, Freeman-Tukey double arcsine transformations or generalized linear mixed models? If you did not transform the proportion, please, justify.

RESPONSE: We did not use the proportion of reads to infer relative species abundance in our analyses. Read numbers were transformed into occurrence data (1/0) per OTU per site (L. 666). We also mention this in the Discussion, and the recent development of some promising approaches like "spike-in" to infer abundance from read counts (see Yinqiu Ji *et al.* 2019). (L. 483-485).

5) Please, upload the resulting metabarcodes (final Operational Taxonomic Units; OTU) to the Barcode of Life Datasystems (BOLD).

RESPONSE: We think BOLD is not a suitable repository for our metabarcodes because our data set does not include morphological vouchers (~3000 MOTUs). However, in line with comment 5 from referee 2, and comment 1 from referee 4, we updated the Supplementary Table I to make the consensus sequence for each MOTU available to the reader. Raw sequencing data are deposited and publicly available in GenBank (SRA accession PRJNA702908).

6) *Are there any endangered insects in the study area (montane Pyrenean forest)? If yes, do drought-induced forest dieback and subsequent salvage logging have an impact on the endangered insects?*

RESPONSE: Regarding the presence of endangered species in our study we checked the MOTU taxonomic names for Lepidoptera for which we have a comprehensive understanding against the IUCN Red list and did not find any species.

7) *Line 45, consider adding "the" between "explain change".*

RESPONSE: Corrected accordingly.

8) *Line 178, consider changing "influences" to "influence".*

RESPONSE: Corrected accordingly.

9) *Line 315, please, explain the artefact of the use of Malaise traps. For example (i.e., sample mainly flying insects).*

RESPONSE: Corrected accordingly.

10) *Line 978, Itallize the species name "Aphidius ervi"*

RESPONSE: Corrected accordingly.

11) *Line 911, 939, 966, 969, 991, 997, 1061, 1065, 1071, 1074, 1085, 1096, 1129, 1158 and 1178. Other than the first word with a capital letter, other words should be in small letters. For example Line 911, change "How Many Species of Insects and Other Terrestrial Arthropods Are There on Earth" to "How many species of insects and other terrestrial arthropods are there on Earth? ".*

RESPONSE: Corrected accordingly.

12) *Remove the additional space on Line 370, 443, 544, 586 and 613.*

RESPONSE: Corrected accordingly.

13) *Line 326. There is only one study cited here but the authors wrote "studies". Either change the word "studies" to "study" or cite more articles.*

RESPONSE: We corrected “similar to other studies¹³” into “similar to a meta-analysis¹³”

Referee #2:

1) First, the paper is strictly based on incidence (rather than abundance) data. While there is little scope of doing differently with meta-barcoding data, it should at least be discussed (and preferably: explored!) how robust the results are in relation to the massive sampling problems that can arise from using incidence data with arthropods. For example, the number of species in common between samples, which is central to ‘zeta diversity analysis’, obviously can strongly depend on the completeness (and size) of samples. ‘True’ co-occurrences are notoriously under-rated with incidence data, because ‘unseen’ species are not necessarily ‘absent’. See the many developments by Rob Colwell, Anne Chao and colleagues (like the Chao-Soerensen or Chao-Jaccard similarity measures, or also the NESS/CNESS measures), over the past 20-30 years in dealing with the many singletons and doubletons in biodiversity inventories of species-rich arthropods! At the very least, the readers should learn about sample sizes. Trap catches of arthropods may easily vary by 1-2 orders of magnitude between samples. Hence, it should be reported how large samples were, and whether sample sizes (e.g. dry mass of bulk samples?) systematically co-varied with forest characters or intensity of salvage logging. From your Figure 2 it is obvious that for all relevant insect orders, even at the landscape level, species richness was NOT exhaustively covered (the species accumulation curves are all still increasing well). Hence, at the level of individual samples coverage will be even MUCH lower, and this does affect your ‘zeta estimates’ (or in other words: you inflate/exaggerate ‘rarity’ just due to species missing/unseen in samples, viz. due to false negatives).

RESPONSE: The referee raises several major points, please see below our answers:

- a) *completeness of sampling, sample sizes & sampling curves* – Malaise trapping comes with biases against certain groups like Lepidoptera and Coleoptera that would have been better represented with the use of light-trapping or flight-interception trapping methods. However, we clearly state the analyses to be on “Malaise-trappable diversity” and discuss such biases in sampling efficiency L. 173, 179, 339-340, 401, 476, 490. Nevertheless, variation is inherent in many aspects, like Malaise trapping positioning as well, regardless of the environment (see Steinke *et al.* 2021). Along with the already discussed species composition bias from the method itself, these cannot be fully controlled in non-experimental plots and may be extended to any biodiversity study with Malaise trap, and most likely to any passive sampling methods in general.

We also agree that accumulation curves still increase, but do approach saturation as clarified in the manuscript, with sampling coverage being estimated at nearly ~90% (see L. 173, L. 181). Hence, by doubling the number of samples (*i.e.* ~400), only a few more Malaise-trappable species would be recovered, regardless of the order (see Figure 3B). Additionally, species richness is intrinsically lowered by our very conservative metabarcoding approach (at least 3 reads present in all the 3 PCR replicates for each retained OTU, see L. 618-641) performed to prioritize the removal of any biological artefact from our dataset (type I error, *i.e.* false positive) to the detriment of more species present on-site or even in the sample but unseen (type II error, *i.e.* false negative). However, this allows us focusing on true (co-) occurrences only and avoid biases from any false ecological signal, critical for future applied forest managements based on those results

- b) *Singletons* – We do account for singletons (*i.e.* species occurring in a single trap only) in our Chao estimator analysis (L. 668). However, we do not weight more singletons in the analyses to avoid inflating the results in expected diversity as our current singletons might be products from molecular analyses biases (DNA extraction, PCR, sequencing...).
- c) *Biomass* – As explained above, trap catch variations may be important with passive trapping methods, but we unfortunately have no biomass data to compare such variation. However, a companion study based on window-flight traps set up in the same plots and focusing on saproxylic beetles did not reveal any significant change in their biomass nor abundance to environmental features or decline level (Cours *et al.* 2021).

2) *Second, I have problems with the mixed usage of the concept of ‘rarity’ in the paper. A species can be ‘rare’ in space, in time, and/or by numbers. Your analytical approach mixes that all up, and does not allow disentangling these very different dimensions of ‘rarity’. Concerning numbers – you effectively have NO DATA on that using meta-barcoding, since you do not have counts or abundances of individual species. Honestly and openly say that, please. Rarity by time could and SHOULD be eliminated from your analysis, because the way you have subjected your data to ‘zeta-analysis’ exaggerates ‘rarity’. Why? Because you used individual trap samples from the same sites, but taken at different times of the year, as putatively ‘independent’ data points. Which they are NOT! To illustrate my point: when you compare the numbers of shared species between samples, across space AND time (as you did) you will inflate the apparent ‘rarity’ due to simple (and massive!) phenological / seasonal species turnover in every temperate-zone ecosystem. For example, cockchafers (*Melolontha*) will only be there in May, so these seasonally very restricted insects (as far as their adult volant stages are concerned) will exaggerate ‘rarity’ because obviously even at the very same place these beetles will not be found in any Malaise trap samples earlier or later at the same site. You could replace *Melolontha* with any other of the very many insects whose winged adults have a short emergence period, often of less than a month per site. However, this problem can easily be solved by aggregating repeated samples from the same site into ONE species list per SITE, and then re-do the analysis. Then, what would remain is one dimension of rarity, *i.e.* rarity by space (= by ecosystem or habitat), and this is the rarity you are talking about in all your paper. So, better configurate the analysis such that it matches your hypotheses and research questions.*

RESPONSE: We thank referee #2 for this important and very valuable comment. First, we agree that temporal replicates should be viewed as pseudo-replication and not independent replicates, and that metabarcoding does not allow rarity inference from read abundance (see response above and comment 4 from referee 1, above).

However, there might be confusion about when we use temporal replicates as different and independent samples and when we use concatenated temporal replicates as a single sample site. As asked by referee #4 in comment 4, we have now clarified in the text accordingly (L. 187-188 and 197). Monthly temporal replicates were used for temporal turnover and accumulation curves analyses only (Fig. 2; 3).

For the remainder of our study, analyses are solely based on concatenated monthly samples as unique sample sites. Hence, zeta-diversity and MS-GDM analyses reflect rarity by space only, for both geographic or dieback/salvage logging conditions. For heatmap analyses, we performed randomized bootstrap on the whole dataset to infer variability in space ability of being Malaise-trapped for each MOTU, hence a “relative” abundance at each given site.

We have now clarified accordingly the used of rarity space dimension with relative abundance in the material and methods (L. 708-716).

With regard to temporal rarity playing a role in our zeta-diversity analyses, we did investigate with non-pooled pseudo-replicates. Results clearly show the issue emphasized as most of the Zeta ratio curves (both for different order and the general one) drop to 0 indicating no common species. A similar pattern is visible for the functional groups (see figures below). Hence, our concatenation of monthly replicates does provide a general and corrected picture of the presence of the MOTUs in a given environment. Comment 3 from referee 4 also discuss the occurrence of MOTUs as a level of time elapse and geographic distance and the subsequent generated heatmap highlight such discrepancies between sampling months or areas overall, and the corrected occurrences from concatenated sampling periods.

NN Zeta ratio with Monte Carlo chain with 5000 iterations for most represented orders through dieback gradient and over time (May, June, July, August)

NN Zeta ratio with Monte Carlo chain with 5000 iterations for the four functional groups through dieback gradient and over time (May, June, July, August)

3) Third, it was unclear to me how exactly you assigned MOTUs to functional categories. Was that based on the ecological functions of the larval or adult stages? Was an ichneumonid wasp classified as a parasitoid (based on larval resource use) or as a nectarivore (based on adult feeding habits)? This applies to most holometabolous insects, notably Diptera, Hymenoptera and Lepidoptera, where the larvae and adults often occupy entirely different feeding niches. It must be explicitly stated whether your functional classifications refer to larvae or adults. Also you noted that you used insect families as proxy for scoring functions. I really disagree with that and use one simple example to illustrate my point, the Lepidopteran family Erebidae. Larvae of these may either be herbivores, detritivores, or fungivores. Adults may be nectarivores, may visit rotting fruits (Catocalinae), or may not feed at all (Lymantriinae, most European Arctiini). How, then, can you use family-level assortments for further functional analysis? Similar problems show up in many other more speciose insect families, e.g. Syrphidae in the Diptera, Formicidae in the Hymenoptera, etc.

RESPONSE: We first assigned each of the 258 insect families recovered to different trophic guilds (*i.e.* saproxylic, zoophagous...) based on the known ecological functions of both immature and adult stages. Then, each family was assigned to four main functional groups only: floricolous / non floricolous adult guilds and parasitoid / non-parasitoid larval guilds. This was done based on the capability of a given MOTU to function as a floricolous species and/or parasitoid (or none) at least over one part of its life (therefore fulfilling the given function in the ecosystem overall). We have now clarified L. 652-659 accordingly. However, feeding guilds have been subsequently removed from Supplementary Table II for better clarity and to avoid confusion for the reader as they were not used in the subsequent functional analyses.

Therefore, to ask your questions:

- “Was an ichneumonid wasp classified as a parasitoid (based on larval resource use) or as a nectarivore (based on adult feeding habits)?” Ichneumonids were classified both as floricolous and parasitoids (see Supplementary Table II).
- “How, then, can you use family-level assortments for further functional analysis?” Indeed, the hugely diverse family of Erebidae shows a variety of different trophic strategies but no Erebidae or Lepidoptera for that matter are known to be parasitoids so we assigned them as floricolous since some Erebrids are known to be pollinators.

We fully agree that more precise ecological and functional studies would require a genus or even species-specific classification. However, with that many data and dark taxa, this would be virtually impossible and can currently be performed only for those MOTUs which have been identified down to species or genus level. This is why we did not use those finer traits and identified trophic guilds to perform more analyses. While this should not be viewed as a shortcoming impeding such analyses, it does imply a tremendous work to screen the literature. This issue also highlights the lack of global and curated database on functional traits for insects.

4) Fourth, I would be eager to know how you have dealt with the many inconsistencies that one often encounters when working with the BOLD barcoding data base. The same insect species may show up there under different names, or there often is a mismatch found between taxonomic ‘species affiliations’ and so-called BINs in BOLD. Did you really not encounter any such inconsistencies when allocating your samples into ‘species’? You obviously also, from understandable pragmatic reasons, used simplified thresholds of sequence similarity to delineate species, genera, etc. Many studies on barcoding with animals have, however, shown

that a universal “barcoding gap” does not exist. Hence, it would be important to use your own large multi-species data to support whether, for example, the 97% similarity threshold (i.e. 3% divergence) at the species level is really valid, across all insects sampled in your study. Or whether, perhaps, a higher threshold might be more meaningful for insect order A, and a lower one for insect order B. Would your results change substantially if you employ, for example, a 2% or a 5% divergence threshold at the species level?

RESPONSE: We agree that no universal barcode gap exists, but used the commonly applied 97% similarity threshold for COI among insects (see Hebert *et al.* 2003 that highlighted intraspecific divergence to be rarely above 2%) as it has often been proven reliable and allows for comparability with other studies. There are indeed studies showing variability among insect Orders or Families (*e.g.* 2% might be more accurate for some parasitoid wasps with higher AC content and divergence rate). However, lowering the threshold as such based on one group would potentially inflate the number of MOTUs overall. As stated in our reply to comment 1, our conservative approach reduced the number of MOTUs, rather than inflating on recovering more species/subspecies or rare species with potential artefacts and/or false positives. Similarly, following your suggestion of implementing a threshold based on taxonomy, this suppose to have the taxonomy associated to the reads BEFORE the clustering step (while we clustered and then associated taxonomy to MOTUs). This is rather an ESV/ASV approach (valid but our study was not designed as such) which requires more computing time and power in order to have a taxonomy-based list and then cluster the associated sequences into MOTUs at various thresholds. It also requires to have a prior knowledge of the divergence threshold among different taxa. While some groups are well-studied and referenced to do so, many other would require trial-and-error to confirm a given threshold, and no morphological voucher (due to grinding) would have been available to infirm/confirm those potential specific thresholds.

5) For the sake of open science, I also strongly recommend to provide a full species x site matrix in an electronic appendix. Without such a “species list” nobody will ever be in the position to cross-check your results – yet repeatability and reproducibility are essential dimensions of science.

RESPONSE: We have updated Supplementary Table I by adding the DNA consensus sequence for each MOTU. This revised table now also contains occurrence data for monthly sample replicates for better repeatability. See also our replies to comment 5 from referee 1, and comment 1 from referee 4.

6) L 69: too hot summer temperatures can also be harmful for insects (‘heat stress’)!

RESPONSE: Added (L. 69-70).

7) L 130: how reliable is family-based classification; especially in the light of your own wording above in the manuscript where you state that species-specific responses to environmental change may predominate? See also my doubts above on the validity of family-level generalizations.

RESPONSE: The analysis of functional groups at family level can be reliable and most families have relatively conserved functions (*i.e.* parasitoids, floricolous...) (Sanderson Bellamy *et al.* 2018). In addition, the assignment of MOTUS to family level is also highly reliable. Indeed, in our experience in DNA barcoding campaigns (*i.e.* saproxylic beetles, leaf-

mining insects, bees...) BOLD allows to classify most records at family level, only in tropical faunas you might find the odd record which cannot be assigned to family level. We clarified this discussion L. 430-432.

8) *Fig. 4: please say explicitly which ordination method you have used to create these diagrams. If this was an eigenvalue method: please report the eigenvalues (or explained variance, or fraction of trace of the relevant matrices, or whatever) at the coordinate axes.*

RESPONSE: Ordination plots were created using the Gaussian copula method implemented in the *ecoCopula* R package (now clarified in the caption L. 880). As such, x and y represent ordination axes 1 and 2, the two latent variables, also called factor 1 and factor 2.

Fig. 7: what precisely is the test statistics that you report here to take values between -0.08 and +0.08? I did not get that, perhaps I overlooked it – yet this must be clearly reported in the figure legend (which should be self-explanatory). Are these (standardized?) correlation coefficients? If so, the pertinent effect sizes would appear to be minute to me.

RESPONSE: The heatmaps represent the *traitGLM* Gaussian copula ordinations, deriving from a fourth corner model fitted with a GLM and LASSO penalty that restricts the model to the most informative interactions only. Thus, the values reported are not standardized, independent from effect size but do represent the most extreme (and significant) values of each generated model. We specified in the figure caption that LASSO penalty was applied to the model (L. 942).

Referee #3:

1) There are no control stands, i.e. stands where the tree decline associated with the drought event of 2003 was not present.

RESPONSE: We believe the naming of our tree dieback categories may be misleading, because stands where tree decline was not present were used as our negative control. Indeed, plots referred to as “low” underwent drought events since 2003 but expressed little to no dieback, similarly to plots that have not been affected by tree diebacks associated from drought events or drought events *per se*. We have now clarified this issue with the text L. 514-515 and 538-539.

2) The hypotheses are not clearly formulated or added at later time, i.e. when the results were obtained.

RESPONSE: We rephrased hypotheses to better reflect the questioning, research development and scheme of our study, especially the use of zeta-diversity analyses based on our prime hypothesis of no change in species richness (L. 139-144).

3) The time span is limited to one year of observations

RESPONSE: The time span of our study is indeed limited to one year but does not impede the inference of ecological variations along the dieback gradient and salvage conditions, because all plots were affected by the same 2003 drought event. The field sites are also close enough that any anomalies of weather that occurred during the study year would have occurred at all sites. However, this study now allows to have a new and more comprehensive “T=0” picture of the biodiversity at those given areas and may serve for future monitoring of ecosystem recovery and biodiversity changes over time. We now mention this in the text (Lines 493-496).

Referee #4:

1) L154: here, you mentioned 224 (=56 plots x 4 months) Malaise trap samples. However, in Suppl Table 1 you only reported 56 sample plots. I would suggest that you release the occurrence matrix for the 2972 MOTUs in 224 traps. This will allow readers to repeat your analyses. I also consider the complete dataset itself a valuable asset to the research field.

RESPONSE: We have now updated Supplementary Table I showing the monthly occurrence matrix for each MOTU including the consensus sequences, in line also with comment 5 from referee 1, and comment 5 from referee 2. It also helped us correcting the actual number of samples processed. 222 samples out of the 224 expected samples were processed in the wet-lab (2 were lost), 211 samples were successfully sequenced and used in the statistical analyses after final demultiplexing and PCR replicates concatenation. We corrected throughout the text accordingly (L. 37; 132; 163-164; 167; 545; 615; 831).

2) L155, 'iNEXT extrapolation estimation...': I understand that you mentioned the specific method used for richness estimation in the Methods section. It is still necessary to briefly mention here the method you used in iNEXT R package.

RESPONSE: We clarified iNEXT parameters accordingly (L. 170).

3) L167-176: You are discussing the patterns of distance decay of similarity, as well as the similarity decay with time elapses. As an option, you might consider to present the level of similarity as a function of geographic distance and time elapse.

RESPONSE: Fixing time and distance together in MSGDM would be biased as change from time decay according to our sampling design relies on temporal pseudoreplicates (one trap per plot sampled four times), and would be inherently significant in regards to the short phenology of insects. We agree that our approach subsequently reduces the contribution of rare species on a temporal basis, but we argue that it reflects a more conservative approach that helps disentangle both geographic and ecological effects on community changes.

In further considering to this comment, we constructed heatmaps depicting the occurrence of each MOTU in each month for each valley (left panel: Aure in light brown; center panel: Sault in dark brown; R1, R2, R3 and R4 correspond to May, June, July and August): We also constructed a heatmap depicting the occurrence of each MOTU throughout the whole period (right panel) in each valley, and MOTUs that were found in both areas (seagreen). Plot will be included with other reviewing figures in a dedicated folder of the GitHub repository or can be included as a supplementary material if necessary.

4) L178: Perhaps clarify from the section you started using pooled monthly data for only 56 plots.

RESPONSE: We clarified the sample nature L. 187-188 and L. 197 as suggested.

5) For the zeta diversity calculation and analyses, I presume that you used Monte Carlo sampling ("mc") instead of Exact method ('ex') due to the extremely large numbers of possible combinations for 56 plots. Please specify the number of site combinations used for calculating zeta diversity and also the MS-GDM.

RESPONSE: The referee was correct. We have now re-run all zeta diversity analyses and corrected supplementary figures and used Monte Carlo sampling (mc) for both the NN scheme (figure 4, 6 and 7) and (for the ALL scheme (figure S4, S5 and S6). All zeta and MSGDM analyses were run with a parameter sample set to 5000. We clarified accordingly L. 739-740 and 756-757 and in all figure captions.

6) Fig.4: The meanings of these clustered black lines in the centre of the plot should be provided in the legend.

RESPONSE: Each black line represents the resultant of both factors 1 and 2 of the traitGLM Gaussian copula ordination for each studied plot reduced to 95% of the 2.5 set alpha-ratio. We clarified in the figure caption accordingly (L. 883-885).

7) In the zeta ratio plots, I noticed that there are a few retention curves have reached zero at high zeta orders, while others didn't (e.g. Fig.5). Please specify whether you have removed sites without the focal taxa in the calculation. I'm not suggesting that whether this practice is correct or not. Instead, they lead to different interpretations of the results (only slightly). So, it's better that you can specify it in the text.

RESPONSE: Calculation were performed without removing any site, including those without focal taxa. We clarified the text accordingly (L. 751-753).

8) L269: This section discusses the I-spline response curves of different predictors. The figures are insightful and contain quite a large amount of information. Besides the magnitude of a curve reflecting the importance (contribution) of a predictor, the specific shape (especially the slope at specific values of a predictor) also include detailed information: a greater slope is indicative of higher sensitivity of zeta compositional turnover to changes of the predictor at a specific value. For instance, compositional turnover is highly sensitive when canopy openness is less than 0.2 (rescaled value) but not so much when canopy openness is high (>0.2). I hope that the authors can look into such details in Fig.8 and add such details for each key predictors.

RESPONSE: Corresponding results (L. 294-307) and discussion (L. 452-464) sections were improved accordingly.

9) For the MS-GDM, it is also important to point out which metrics were used. I suspect that the authors used the raw zeta diversity. There are also different zeta metrics to choose (e.g. normalised, standardised, Sorensen-equivalent, Simpson-equivalent), with each providing slightly different meanings. In addition, if the model used a specific number of combinations for each zeta order, I would prefer that you run the model multiple rounds to check whether

the I-spline response curves of predictors are stable or not. You can have a look of Latombe et al. (2019) Journal of Biogeography 46(10): 2299-2310.

RESPONSE: MS-GDM analysis and Fig. 9 panels are based on Sørensen-equivalent metric, with very small standard deviation of each zeta diversity variance explanations (<0.001). We clarified the text accordingly (L. 761-763; 958-959). In addition, MS-GDM analysis was run over 30 rounds for stable I-spline response curves of predictors, with corresponding Fig. 9 representing the average value of these 30 rounds.

We provide below two other MS-GDM recalculations, both based on Simpson-equivalent or Jaccard-equivalent metrics, respectively. Because conclusions are very similar among the three different models, we kept “Sørensen” in the text.

MS-GDM recalculation based on Simpson-equivalent metric:

MS-GDM recalculation based on Jaccard-equivalent metric:

All newly generated panels are now available on GitHub https://github.com/Lucasire/Malaise_FR_2017, in a .zip folder named “4_Reviewing – Additional figures” along with the scripts of the article.

References

- Cours, Jérémy, Laurent Larrieu, Carlos Lopez-Vaamonde, Jörg Müller, Guillem Parmain, Simon Thorn & Christophe Bouget.** (2021). 'Contrasting responses of habitat conditions and insect biodiversity to pest- or climate-induced dieback in coniferous mountain forests'. *For. Ecol. Manag.* **482**, 11881, 1–14. <https://doi.org/10.1016/j.foreco.2020.118811>
- Hebert, Paul D. N., Sujeevan Ratnasingham & Jeremy R. deWaard.** (2003). 'Barcoding animal life: cytochrome *c* oxidase subunit 1 divergences among closely related species'. *Proc R. Soc. Lond. B.* **270**: S96–S99. <https://doi.org/10.1098/rsbl.2003.0025>.
- Ji, Yinqiu, Tea Huotari, Tomas Roslin, Niels Martin Schmidt, Jiaxin Wang, Douglas W. Yu & Otso Ovaskainen.** (2019). 'SPIKEPIPE: A Metagenomic Pipeline for the Accurate Quantification of Eukaryotic Species Occurrences and Intraspecific Abundance Change Using DNA Barcodes or Mitogenomes'. *Molecular Ecology Resources*: 1–12. <https://doi.org/10.1111/1755-0998.13057>.
- Latombe, Guillaume, Núria Roura-Pascual & Cang Hui.** (2019). 'Similar compositional turnover but distinct insular environmental and geographical drivers of native and exotic ants in two oceans'. *Journal of Biogeography* **46**(10): 2299–2310. <https://doi.org/10.1111/jbi.13671>.
- Sanderson Bellamy, Angelina, Ola Svensson, Paul J. van den Brink, Jonas gunnarsson & Michael Tedengren.** (2018) 'Insect community composition and functional roles along a tropical agricultural production gradient'. *Environ. Sci. Pollut. Res.* **25**: 13426–13438. <https://doi.org/10.1007/s11356-018-1818-4>.
- Steinke, Dirk, Thomas W. A. Braukmann, Laura Manerus, Allan Woodhouse & Vasco Elbrecht.** (2021) 'Effects of Malaise Trap Spacing on Species Richness and Composition of Terrestrial Arthropod Bulk Samples'. *Metabarcoding and Metagenomics* **5**: e59201. <https://doi.org/10.3897/mbmg.5.59201>.

REVIEWERS' COMMENTS:

Reviewer #1 (Remarks to the Author):

The authors revised the manuscript following all comments and suggested changes. All comments/suggestions were considered, and I am happy to report that all of them led to at least some improvement in the manuscript.

Minor comments/recommendations:

- 1) Line 166, consider changing "specie-level" to "species-level"
- 2) Line 440, consider adding "To" in front of "Investigating ..."

Reviewer #2 (Remarks to the Author):

The very numerous queries and suggestions from all previous reviewers have been tackled in a manner that I feel is satisfactory. Many of the earlier comments were based on somehow unclear usage of wording, or on a lack of detail with regard to some of the (highly sophisticated!) means of data analysis.

Admittedly I am NOT an expert in most of these statistical techniques. But to me the paper now makes very good reading and demonstrates that (not really unexpected!) the compared forest stands show no consistent variation in species richness (of arthropods amenable to study through Malaise trap samples), but reveal massive change in functional composition. The study also shows that Diptera and Hymenoptera (that can hardly be studied at species level using "classical" taxonomic information; "dark taxa") were particularly sensitive to differences in environmental conditions. Hence, the use of meta-barcoding is clearly advocated in such cases, and the present version of the paper also addresses adequately the opportunities and limitations of that approach. Also the limitations of Malaise trap samples are now more explicitly stated than was the case with the initial submission.

Reviewer #3 (Remarks to the Author):

Question 1

The authors have explained what they mean as control, i.e. trees where damage was <60% and explain at lines 514-515 that these trees acted as a healthy control. This is not acceptable as a healthy tree is generally expected to have <10% of damage.

Question 2

The hypotheses have been modified but in relation to unresolved question 1, they are still not clear.

Question 3

I am fine with the justification provided.

Reviewer #4 (Remarks to the Author):

I have carefully assessed the authors' response to my previous comments and the revised manuscript. All comments have been adequately addressed in this revision, and I appreciate the precision in the authors' response. I have no further comments.

Communications Biology submission 21-1178-A

Authors' 2nd response to reviewers and editor

Summary

Global comments	2
Referee #1 :.....	3
Referee #3 :.....	4

General comments

We have made the following major changes to our manuscript following editor's request and to improve clarity:

- MSGDM analyses and Figure 9 were re-made adding TreM density.
 - o "Data not shown" statement has now been removed.
 - o Manuscript has been modified accordingly (L. 151–152; 295–311; 457–468; 522–523).
- Heatmap color representation has been defined on the Figure 8.
- "Statistical analyses" section has been changed for "Statistics and Reproducibility".
- "Data availability" section has been changed for "Data and Code availability".
- "Conflict of interest" section has been changed for "Competing interests".
- Scripts and associated datasets have been publicly released on Zenodo.
 - o A DOI is now provided in the "Data and Code Availability" section.
 - o Associated DOI has been cited as requested (reference n°96).
- Supplementary Table I and II as well as Supplementary Information – Plot and Primer lists have been renamed as Supplementary Data I–IV and deposited for public release on Figshare.
 - o In-text labels have been corrected accordingly.
 - o A DOI is now provided in the "Data and Code Availability" section.
 - o Associated DOI has been cited as requested (reference n°97).
- Raw sequencing data are now publicly available on GenBank.
- Model equations are now numbered sequentially.

Referee #1:

1) Line 166, consider changing “specie-level” to “species-level”

RESPONSE: Corrected accordingly.

2) Line 440, consider adding “To” in front of “investigating...”

RESPONSE: Corrected as “To investigate...”.

Referee #3:

1) The authors have explained what they mean as control, i.e. trees where damage was <60% and explain at lines 514-515 that these trees acted as a healthy control.

RESPONSE: Our previous description did not mean “trees where damage was <60%” which would mean all trees being affected, at least to a small extent. Instead, we meant that at least 40% (or more) of the trees were healthy or resilient. We rephrased accordingly L. 514–515 and also explained that fully healthy stands are difficult to find in a dieback context, nuancing that our stands acted as “healthier” controls (L. 517).

2) The hypotheses have been modified but in relation to unresolved question 1, they are still not clear.

RESPONSE: We believe that our rephrasing on healthy stands (*question 1* above) makes the hypotheses clearer.